# Altimeter Calibrations in the Preliminary Four Years' Operation of Wanshan Calibration Site

Wanlin Zhai [1], Jianhua Zhu [1], Hailong Peng [2], Chuntao Chen [3,*], Longhao Yan [1], He Wang [1], Xiaoqi Huang [1], Wu Zhou [2], Hai Guo [1] and Yufei Zhang [2]

1   National Ocean Technology Center, Tianjin 300112, China; zwl13032@163.com (W.Z.); besmile@263.net (J.Z.); reed1984@163.com (L.Y.); wanghe_sio@126.com (H.W.); 13920046686@163.com (X.H.); g.h1985@163.com (H.G.)
2   National Satellite Ocean Application Service, Beijing 100081, China; phl@mail.nsoas.org.cn (H.P.); zhouwu@mail.nsoas.org.cn (W.Z.); lora@mail.nsoas.org.cn (Y.Z.)
3   School of Ocean, Yantai University, Yantai 266004, China
*   Correspondence: chenchuntao@ytu.edu.cn; Tel.: +86-0535-6893172

**Abstract:** In order to accomplish the calibration and validation (Cal/Val) of altimeters, the Wanshan calibration site (WSCS) has been used as a calibration site for satellite altimeters since its completion in August 2019. In this paper, we introduced the WSCS and the dedicated equipment including permanent GNSS reference stations (PGSs), acoustic tide gauges (ATGs), and dedicated GNSS buoys (DGB), etc. placed on Zhi'wan, Wai'ling'ding, Dan'gan, and Miao'Wan islands of the WSCS. The PGSs data of Zhi'wan and Wai'ling'ding islands were processed and analyzed using the GAMIT/GLOBK (Version 10.7) and Hector (Version 1.9) software to define the datum for Cal/Val of altimeters in WSCS. The DGB was used to transfer the datum from the PGSs to the ATGs of Zhi'wan, Wai'ling'ding, and Dan'gan islands. Separately, the tidal and mean sea surface (MSS) corrections are needed in the Cal/Val of altimeters. We evaluated the global/regional tide models of FES2014, HAMTIDE12, DTU16, NAO99jb, GOT4.10, and EOT20 using the three in situ tide gauge data of WSCS and Hong Kong tide gauge data (No. B329) derived from the Global Sea Level Observing System. The HAMTIDE12 tide model was chosen to be the most accurate one to maintain the tidal difference between the locations of the ATGs and the altimeter footprints. To establish the sea surface connections between the ATGs and the altimeter footprints, a GPS towing body and a highly accurate ship-based SSH measurement system (HASMS) were used to measure the sea surface of this area in 2018 and 2022, respectively. The global/regional mean sea surface (MSS) models of DTU 2021, EGM 2008 (mean dynamic topography minus by CLS_MDT_2018), and CLS2015 were accurately evaluated using the in situ measured data and HY-2A altimeter, and the CLS2015 MSS model was used for Cal/Val of altimeters in WSCS. The data collected by the equipment of WSCS, related auxiliary models mentioned above, and the sea level data of the hydrological station placed on Dan'gan island were used to accomplish the Cal/Val of HY-2B, HY-2C, Jason-3, and Sentinel-3A (S3A) altimeters. The bias of HY-2B (Pass No. 375) was $-16.7 \pm 45.2$ mm, with a drift of 0.5 mm/year. The HY-2C biases were $-18.9 \pm 48.0$ mm with drifts of 0.0 mm/year and $-5.6 \pm 49.3$ mm with $-0.3$ mm/year drifts for Pass No. 170 and 185, respectively. The Jason-3 bias was $-4.1 \pm 78.7$ mm for Pass No. 153 and $-25.8 \pm 85.5$ mm for Pass No. 012 after it has changed its orbits since April 2022, respectively. The biases of S3A were determined to be $-16.5 \pm 46.3$ mm with a drift of $-0.6$ mm/year and $-9.8 \pm 30.1$ mm with a drift of 0.5 mm/year for Pass No. 260 and 309, respectively. The calibration results show that the WSCS can commercialize the satellite altimeter calibration. We also discussed the calibration potential for a wide swath satellite altimeter of WSCS.

**Keywords:** satellite altimeter; Wanshan calibration site; calibration and validation; permanent GNSS station; tide gauge; mean sea surface; tide model; dedicated GNSS buoy

## 1. Introduction

Satellite altimeters provide accurate sea surface height (SSH), wind speed (WS), and significant wave height (SWH) measurements [1]. Such measurements are great contributions for technology and research of oceans and inland waters by providing reliable observations of global oceans, sea ice, ice sheets, lakes, and rivers [2]. The time series of accurate altimeter data covers more than 35 years, starting with TOPEX/POSEIDON (T/P), and presently supported satellites include Jason-3/CS, SARAL/AltiKa, Sentinel-3A/B, HY-2B/C/D, CryoSat-2, and Surface Water and Ocean Topography (SWOT). The requirements of the accuracy and stability of the altimeter system are rather stringent for its data applications. The Global Climate Observing System (GCOS) stated that to address questions of climate science, the sea level records at scales of 50~100 km should have an accuracy of order 10 mm, with the accuracy of the global average (on a weekly or 10-day basis) being 2~4 mm and with a stability of 0.3 mm/year. Therefore, calibration and validation (Cal/Val) of satellite altimeters are crucial [3].

Although there are often major Cal/Val campaigns at the start of a satellite mission, it is also essential that such work continues throughout the lifetime of the spacecraft, because an instrument's behavior may change due to continued usage and long space exposure. Furthermore, long-term validation series are not only important for reducing the uncertainty in any bias, but also for allowing the estimation of gradual drifts in the measurements of the altimeters.

To accomplish the Cal/Val of the altimeters in different regions around the world to comprehensively evaluate their performance and keep the consistency and continuity of the measurements, scientists have established several calibration sites, including Lampedusa in Italy [4], Senetosa in southwest Corsica in France and Ibiza Island in Spain [5], Harvest Oil Platform in America [6], Bass Strait in Austrilia, Gavdos in Greece [7], Kavaratti in India [8], and Wanshan calibration site (WSCS) in China (Figure 1). Dedicated infrastructure such as tide gauges, permanent GNSS stations (PGSs), dedicated GNSS buoys (DGB), meteorological instruments, and auxiliary data/models are employed in such sites. Repeat Cal/Val campaigns have been performed in these sites for long-term exposure. Also, the lake Issykkul in Kyrgyzstan has been identified as a suitable location for calibration of water surface height of altimeters over land water bodies. As well as these in situ sites, many campaigns have been conducted to evaluate the performance of the altimeters (blue dots in Figure 1). The scientists also use globally or regionally distributed tidal stations to assess the relative accuracy of the altimeters.

The preliminary task for the Cal/Val of altimeters was to set the unified datum of the site or campaign. This is achieved using the GNSS data of several PGSs placed in the calibration sites or campaigns [4–8]. The GNSS data make combined progress with nearby GNSS stations using GAMIT/GLOBK, Berenese, GIPSY, or precise point positioning (PPP) software or methods. The accuracy of the height datum can reach to less than 5 mm [9,10].

After that, there are many methods to accomplish the Cal/Val of altimeters, such as the DGB, tide gauges, transponder, moored oceanographic instruments, etc.; the tide gauges are the most commonly used method [5,7,9,11]. In this method, the absolute datum of the tide gauge should be determined using the PGSs plus leveling, kinematic GNSS measurements, or DGBs [12]. Moreover, the tide gauge is often placed on land/an island, while the altimeter signals are contaminated in such an area [11]. There is a distance of about 15~30 km from the tide gauge location to the non-contaminated altimeter footprints. The mean sea surface (MSS) and tidal corrections must be taken into consideration [11,12]. The MSS gradient can reach up to 10 cm/km in some places, which is the key correction element for altimeter calibration [1]. Many global MSS models developed by Technical University of Denmark (DTU), Collecte Localisation Satellites of Centre National d'Etudes Spatiales (CNES_CLS), Shandong University of Science and Technology (SUST2020), and Wuhan University (WHU) were constructed using the altimeter data, tide gauge data, and other auxiliary data [13,14]. For tidal corrections, the more precise global or regional tide models were used in the Cal/Val of altimeters such as FES2014, NAO99jb, GOT, etc. [15].

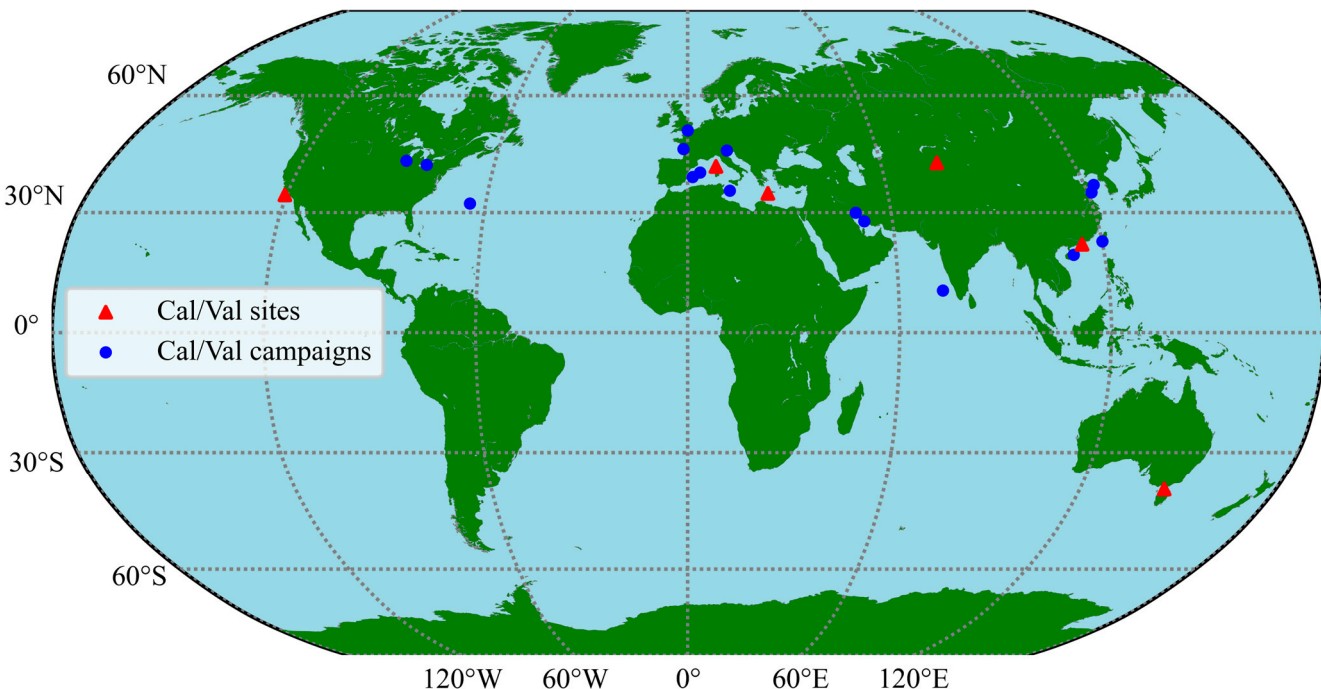

**Figure 1.** Satellite altimeter calibration sites and experiments. The red triangulars represent the operational in situ calibration sites. The blue dots represent the altimeter calibration campaigns.

The WSCS has been chosen to be the first specified site for Cal/Val of the satellite altimeters in China since 2013 by the National Satellite Ocean Application Service (NSOAS) [16]. They plan to accomplish Cal/Val for HY-2 series satellites. Its facilities are placed on four islands that are adjacent to the footprint of HY-2A/B/C. Fortunately, the Sentinel-3A (S3A) and Jason-3 altimeters also fly over this site. This makes it possible to calibrate such altimeters. In this research, we aim to assess the accuracy of the above altimeter missions by comparing the SSH with in situ measurements of WSCS. In contrast to previous work in the region [15], our study is more comprehensive and particular. We have introduced the equipment and data processing strategies of WSCS in detail since August 2019. The ocean models such as the MSS and tide were also evaluated and used in this research. Moreover, our research can provide a unified datum for multiple altimeters, and ensure the consistency and continuity of the measurements by long-term series altimeters.

A detailed introduction of the facilities for altimeter calibration in WSCS and the altimeter calibration methods were provided in Section 2. In Section 3, the datum of altimeter calibration has been defined using the PGSs and DBG. Moreover, the related data and correction models for Cal/Val of altimeters such as the MSS and tidal differences of WSCS were measured and evaluated. The calibration results of HY-2B, HY-2C, Jason-3, and S3A altimeters were shown in Section 4. Sections 5 and 6 were the discussions and conclusion, respectively.

## 2. Introduction and Altimeter Calibration Method of WSCS

### 2.1. Introduction of WSCS

The WSCS is located at the south of Guangdong Province, China (114.3°E, 22.0°N) (Figure 2). The climate is subtropical and displays clear seasonal variations. About 5.3 typhoons pass through this area every year, causing great economic losses. The typhoons also create difficulties for the fixation of the facilities on this site. However, the continuity of equipment measurement is good because the firmness and stability was considered carefully at the beginning of the construction.

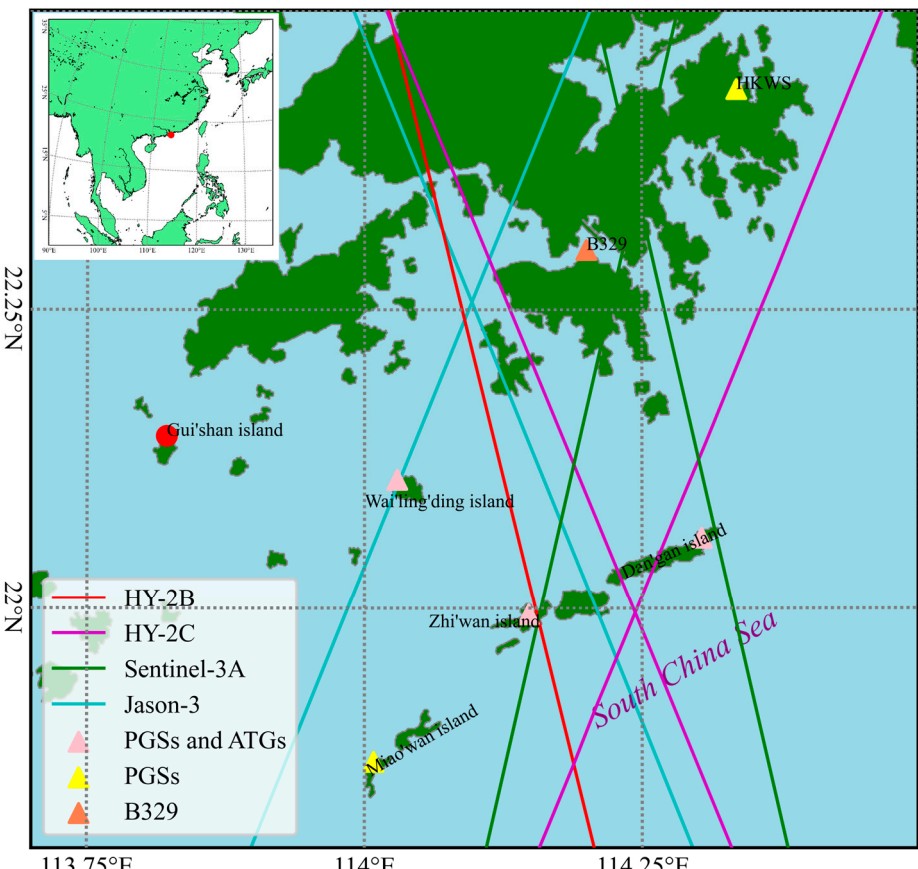

**Figure 2.** Detailed view of the facilities of WSCS and nearby related stations. The pink triangles represent the combined stations of PGSs and ATGs. The yellow triangles represent the station of PGSs on Miao'wan island and Hong Kong (HKWS), respectively. The coral circle represents the laboratory of WSCS on Gui'shan island. The B329 tide gauge station of GCOS in Hong Kong were also included in this research (coral triangular).

The WSCS is a favorable site for altimeter calibrations, as southeast of the Dan'gan, Zhi'wan, and Miao'wan islands extends the South China Sea, and there are no islands to contaminate the altimeter signals (Figure 2). There are four PGSs placed on Wai'ling'ding, Dan'gan, Zhi'wan, and Miao'wan islands, with a data reception period of 1 s [16]. The four PGSs were fixed with the bearing wall of the building or bedrock of the islands to ensure its stability. Each PGS is equipped with a SOUTH Net-S9 receiver and STHCR3-G3 Choke Ring antenna, and the antenna phase center stability is within 2 mm (http://www.ngs.noaa.gov/ANTCAL/, accessed on 10 August 2023). An XZC 6 meteorological instrument and a CE318 sun photometer were placed on the mountain top of Dan'gan island, together with the PGS.

Three acoustic tide gauges (ATGs) were installed with a data reception of 30 s for Dan'gan and Wai'ling'ding islands and 6 min for Zhi'wan island, to measure the sea level, in August 2019. The ATGs are equipped with Aquatrak 5003 sensors, one calibration tube, several extension tubes, and one stainless-steel protective tube with a sea-level measurement accuracy of less than 3 mm. The stainless frame was installed to keep the vertical of the sensor and the tubes and protect them. The DGB was used to define the datum of the ATGs for three times during 2019–2023 [17].

The above devices constitute the main facilities for altimeter calibrations. Moreover, an emergency monitoring center was established for equipment monitoring and maintenance, data collection, and initial processing on Gui'shan island (Figure 2). All the observation data are transmitted to the integration laboratory via BeiDou satellite communication, wireless bridge, and GPRS communication link, and then to NSOAS, China. Moreover, a

moored GNSS buoy, meteorological buoy, and seabed equipped with a pressure tide gauge were also planned to displace about 30~100 km at the southeast of this site.

### 2.2. Altimeters and Calibration Methods

The HY-2B, HY-2C, Jason-3, and S3A altimeters are calibrated in this research, and the SSH of the altimeters ($SSH_{alt}$) are defined using Equation (1).

$$SSH_{alt} = H - (R + WZD + DZD + IC + SSB + SET + LT + PT) \tag{1}$$

where the H is the height of the altimeter above the ellipsoid;
R is the range from the altimeters to the surface of the ocean;
WZD/Dry is the wet/dry zenith delay caused by the wet and dry atmosphere, respectively;
IC is the ionosphere correction;
SSB is the sea state bias which was obtained using empirical models;
SET is the solid Earth tide;
LT is the loading tide height;
PT is the pole tide.

The calibration of the SSH of the altimeters using the ATGs was defined using Equation (2).

$$Bias = SSH_{alt} - SSH_{in-situ} - DMSS - DTidal \tag{2}$$

where the $SSH_{tg}$ is the sea level of the in situ measurements at the time of altimeter overflight which was described in Section 3.2;
DTidal is the tidal differences between the in situ measurements and the altimeter footprints which was described in Section 3.3.
DMSS is the MSS differences between the ATGs and the altimeter footprints, which were described in Section 3.4.

### 3. Related Data and Correction Models in Cal/Val of Altimeters

The datum of altimeter calibration was defined using the four PGSs in WSCS. The height benchmarks of the ATGs were defined using the DGBs with supports of the PGSs. Moreover, the ATGs has a distance of 15~30 km from the altimeter comparison point. Therefore, the MSS and tidal differences must be taken into consideration in such distance [12]. The global/regional tide models were compared using four in situ tide gauge data (Figure 2). A GPS towing-body and a highly accurate ship-based SSH measurement system (HASMS) have been used to measure the sea surface of this area and validate the MSS models [18].

### 3.1. Datum of WSCS by the PGSs

PGS data of Wai'ling'ding (WLDD) and Zhi'wan (ZWAN) islands was collected from 24 August 2019 to 30 June 2023 (ftp://1.203.103.214, accessed on 28 August 2023). These data were processed with a combined solution of 61 GNSS stations around this area, which were downloaded from the GNSS Research Center (GRC), Wuhan University. The GAMIT/GLOBK software was used to process the daily data of the stations, and then the high-precision baseline and coordinate adjustments were obtained. The total number of GNSS data was 60,479, with 1217 data for ZWAN and 1203 data for WLDD. After that, the noise analysis was performed using the Hector software [19,20]. The Generalized Gauss–Markov noise (GGM) and white noise + generalized Gauss–Markov noise (WN + GGM) models were used to analyze the noise of the PGS daily solution data. Details of the PGS processing can be found in [16].

The velocities of the N, E, and U components of ZWAN and WLDD were derived in ITRF 2014 (Table 1). Additional results derived from the Jet Propulsion Laboratory (JPL), Scripps Orbit and Permanent Array Center (SOPAC), and the combined results (COMB) of HKWS, which was only about 30 km from WSCS (ftp://garner.ucsd.edu, accessed on 15 January 2024, Figure 2), was also used to make comparisons with the trends of WLDD

and ZWAN. There was a bit of a difference in the three components (Table 1). This may be caused by the time series of the solution. The SOPAC, JPL, and COMB solutions ranged from June 2010 to July 2024, and the measurements for WSCS take more than three years. This requires longer time series measurements to construct a model of the coordinates. The accurate coordinates of the PGSs will set the datum for the campaign of Cal/Val of the altimeters. The coordinates of the X, Y, and Z components of ZWAN and WLDD were derived in ITRF 2014. Time series of the two PGSs were shown in Figure 3.

**Table 1.** Estimated stations velocities (mm/year).

| PGS | N Slope | E Slope | U Slope | Noise Model | Solutions |
|---|---|---|---|---|---|
| ZWAN | $-10.52 \pm 0.35$ | $30.74 \pm 0.24$ | $-2.05 \pm 0.91$ | GGM | |
| | $-10.52 \pm 0.38$ | $30.74 \pm 0.24$ | $-2.33 \pm 0.80$ | GGM + WN | |
| WLDD | $-10.00 \pm 0.29$ | $31.25 \pm 0.26$ | $-1.73 \pm 0.55$ | GGM | This research |
| | $-10.01 \pm 0.34$ | $31.25 \pm 0.27$ | $-1.76 \pm 0.59$ | GGM + WN | |
| HKWS | $-10.80 \pm 0.31$ | $30.95 \pm 0.21$ | $-1.43 \pm 0.68$ | GGM | |
| | $-10.83 \pm 0.29$ | $30.95 \pm 0.22$ | $-1.59 \pm 0.60$ | GGM + WN | |
| | $-11.54 \pm 0.18$ | $32.57 \pm 0.16$ | $0.73 \pm 0.52$ | -- | SOPAC |
| | $-11.40 \pm 0.09$ | $32.89 \pm 0.14$ | $0.32 \pm 0.30$ | -- | COMB |
| | $-11.40 \pm 0.10$ | $32.94 \pm 0.15$ | $0.07 \pm 0.30$ | -- | JPL |

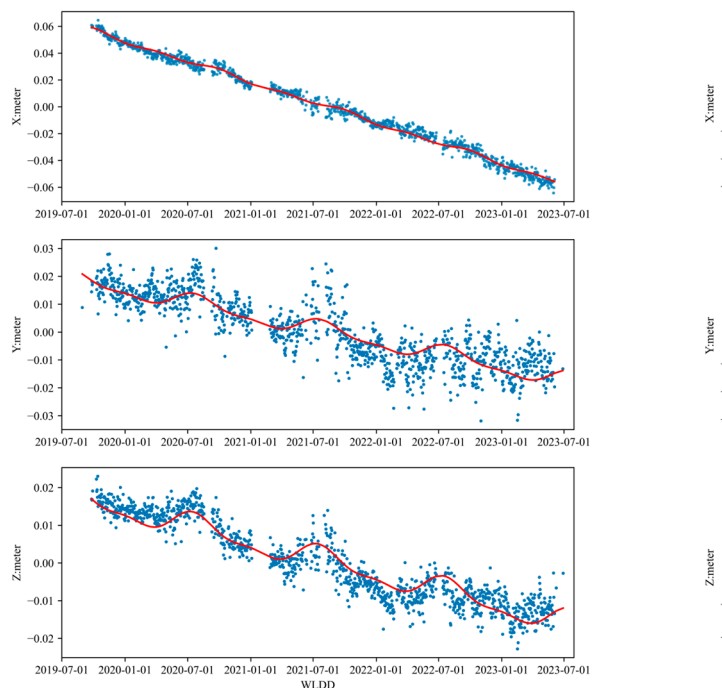
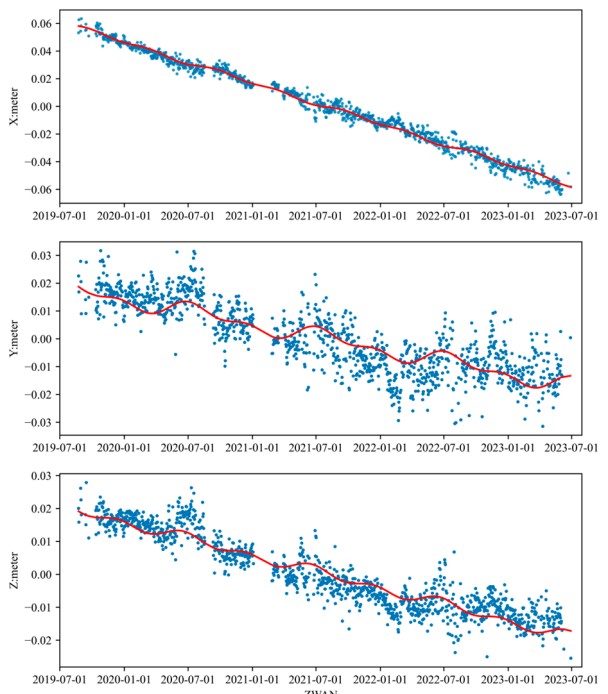

**Figure 3.** Time-series of the coordinates of ZWAN and WLDD. The blue dots represent the raw coordinate processed by GAMIT/GLOBK. The red lines represent the results after removing the noise and geometric modelling.

### 3.2. Datum of the In Situ SSH Measurements

To accomplish the Cal/Val of altimeters, the datum of the three ATGs of WSCS were determined from 2019~2022 using the DGB [17]. The time length of each campaign was more than 3 h for each determination, and the distance between the DGB and the ATGs was less than 300 m. To obtain accurate coordinates, a Trimble Net R9 dual-frequency receiver and Trimble-Ti Chock Ring Antenna were equipped on DGB to decrease the impact of multi-path (MP1/2) effects [17].

The quality of the DGB was checked using the Anubis software (version 3.7) in the campaigns [21]. The data were deleted if the MP1/2 was more than 0.5 m. The received Rinex data of DGB were processed using the accurate coordinates and the static Rinex data of the PGSs of WSCS are needed for the kinematic solutions of DGB. The kinematic DGB data were processed using the TRACK module of the GAMIT/GLOBK together with the final GNSS satellite clocks and orbits [22]. The kinematic results comprised the XYZ, NEU, and BLH coordinates and the integrated weighted RMS of the DGB. A low pass filter of five minutes was used to eliminate the impact of the dynamic ocean environment. Data suddenly affected by the impact will be deleted (Figure 4h). The U/H components of the kinematic solutions were used to set the datum for the ATGs. The overall precision of the SSH can be less than 15 mm. After three years of comparisons, the datum of the ATGs was set with an accuracy of less than 10 mm (Figure 4).

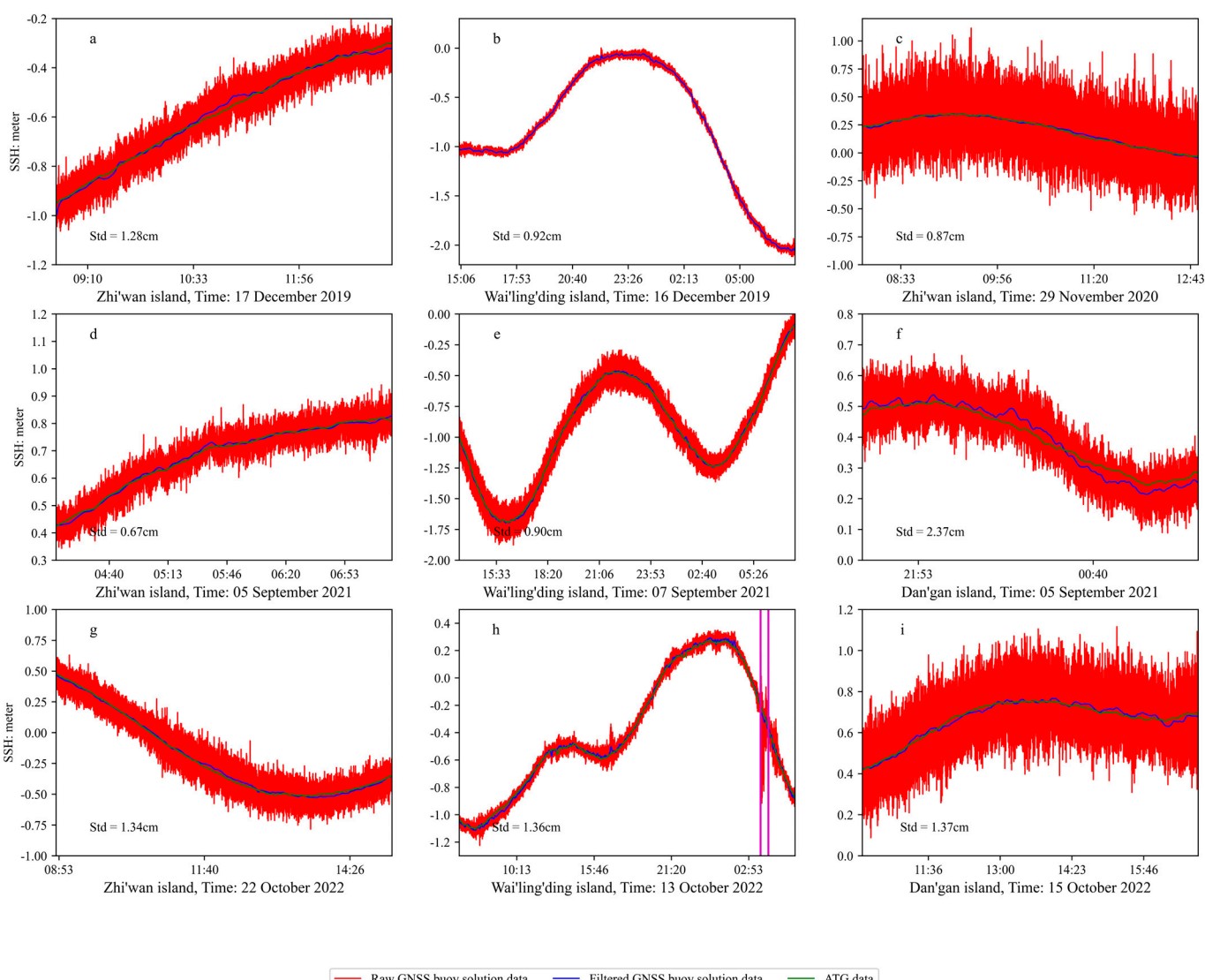

**Figure 4.** Definition of the datum of the ATGs using the DGB. Panels (**a–i**) show the comparisons between the ATGs and the DGB from 2019–2021. The magenta lines in (**h**) represent the SSH of the DGB jumped suddenly. This may be caused by weak GNSS satellite coverage, multipath, or changes in the floating position (Reprinted/adapted with permission from Ref. [19]).

This datum of the PGSs was WGS 84 ellipsoid with an equatorial radius of 6378.137 km and a flattening coefficient of 0.003352811, but have height differences with the S3A, HY-2B,

HY-2C, and J3 altimeter, which have an equatorial radius of 6378.1363 km and a flattening coefficient of 1/298.257 [23]. This height difference was corrected using Equation (3):

$$dh = -W da + \frac{W}{a}(1 - f)sin^2 \varphi df \qquad (3)$$

where, *dh* was the changes of the height caused by reference ellipsoid transformation; *da* = *a*0 − *a*, *df* = *f*0 − *f* were the corrections of the Semi-major Axis and flattening factor, respectively;
*a* and *a*0 were the Semi-major Axis of WGS 84 and altimeter satellite, respectively;
*f* and *f*0 were the Flattening Factor of the Earth of WGS 84 and altimeter satellite, respectively;
$\varphi$ was the latitude, and $W = \sqrt{1 - e^2 sin^2 \varphi}$;
*e* was the first eccentricity of an ellipsoid.

Moreover, the sea level measured by the hydrological station placed on Dan'gan island was collected from January 2017 to December 2019 when the S3A flew through this area (accessed from Pearl River Hydrology and Water Resources Survey Center on 30 March 2020). The distance between the ATG and the hydrological station was less than 5 m and worked at the same time from August to December 2019. They were compared with each other in a standard bias of 13.2 mm. The datum of the data was transformed to be the same with the ATG. Such data were also used to accomplish the Cal/Val of S3A altimeter in this research.

### 3.3. Tidal Differences

The tidal differences in Cal/Val of altimeters were usually determined by global/regional tide models [12]. Here, we collected the tide models of FES2014 (ftp://ftp-access.aviso.altimetry.fr, accessed on 6 July 2023), HAMTIDE 12 (ftp://ftp-icdc.cen.uni-hamburg.de/hamtide/, accessed on 18 February 2023), EOT20 (https://www.seanoe.org/data/00683/79489/, accessed on 9 July 2023), DTU16 (ftp://ftp.space.dtu.dk, accessed on 28 August 2023), NAO99Jb (https://www.miz.nao.ac.jp/staffs/nao99/index_En.html, accessed on 4 September 2019), GOT4.10 (https://earth.gsfc.nasa.gov/geo/data/ocean-tide-models, accessed on 9 July 2023) to maintain the tidal differences (Table 2). The precision of the tide models was evaluated by comparing with the water level data from four in situ tide gauge stations, including three ATGs placed on the three islands of WSCS and one tide station of Hong Kong (No. B329, Figure 2) derived from the Global Sea Level Observing System (GLOSS, http://uhslc.soest.hawaii.edu/data/, accessed on 19 July 2023) [24–28]. The cubic interpolation method was used to obtain the harmonic constants of the ocean tide models of the tide gauge locations. The in situ tidal harmonic constants were analyzed using the t_tide model (version 1.5), which can perform classical harmonic analysis for periods of about 1 year or shorter [29].

**Table 2.** The participating global ocean tide models.

| Tide Model | Facility | Resolution (Degree) | Number of Tidal Constituents |
|---|---|---|---|
| FES2014 | Archiving, Validation, and Interpretation of Satellite Oceanographic (AVISO) | 1/16 | 34 |
| HAMTIDE12 | The Deutsches Geodätisches Forschungsinstitut, Technical University of Munich (DGFI-TUM) | 1/8 | 17 |
| DTU16 | Technical University of Denmark (DTU) | 1/16 | 10 |
| NAO99Jb | National Astronomical Observatory (NAO) | 1/12 | 16 |
| GOT4.10 | National Aeronautics and Space Administration (NASA) | 1/8 | 16 |
| EOT20 | DGFI-TUM | 1/8 | 17 |

The vector differences of the amplitude and phase of tidal components between the in situ data and model predicted values are given by Equation (4) [30–32]:

$$\text{vector} = [\frac{1}{2}(H_{model}\cos(G_{model}) - H\cos(G))^2 + \frac{1}{2}(H_{model}\sin(G_{model}) - H\sin(G))^2]^{1/2} \tag{4}$$

where the $H_{model}$ and $G_{model}$ represent the amplitude and Greenwich phase lag of a tide constituent given by the global tide models, respectively;
The H and G are the amplitude and Greenwich phase lag derived from in situ tide gauges, respectively.

The RMS and RSS differences between the interpolated tidal signal and in situ tide gauge are computed by Equations (5) and (6):

$$\text{RMS} = \langle (\frac{1}{N}\sum_{k=1}^{N}\left(VEC_{tide}^k\right)^2)\rangle^{1/2} \tag{5}$$

$$\text{RSS} = \langle (\frac{1}{8}\sum_{k=1}^{8}\left(RMS^k\right)^2)\rangle^{1/2} \tag{6}$$

where the $VEC_{tide}^k$ is the vector difference at a specific location;
N is the total number of tide gauges;
$RMS^k$ is the RMS of constituent k.

The precision of the six tide models is very close to one another (Table 3), in which the HAMTIDE12 tide model is the most accurate one in this area. The EOT20 tide model has some difficulties in modelling the tides in this region, and the problem occurs in the B329 tide location and K1 tide component. As the parameters of this tide model are not covered the Hong Kong area, the extrapolation is used to obtain the tidal parameters. After excluding the Hong Kong tide gauge, RSS of the GOT4.10, EOT20, and HAMTIDE12 decrease a bit, while the other tide models increase (RSS* in Table 3). Therefore, when using tide models in the Cal/Val of altimeters in WSCS, the GOT4.10 and HAMTIDE12 are recommended. The DTU16, NAO99jb, and FES2014 can also be used carefully by the EOT20 model. In this research, the HAMTIDE12 tide model was used to correct the tidal difference between the ATGs and the footprints of the altimeters.

**Table 3.** Comparisons between the in situ tide and tide model (Unit: cm). RSS include all the four tide gauges. RSS* represent the RSS without the B329 tide gauge (Figure 2).

|  | K1 | K2 | M2 | N2 | S2 | O1 | P1 | Q1 | RSS | RSS* |
|---|---|---|---|---|---|---|---|---|---|---|
| DTU 16 | 2.60 | 1.46 | 9.03 | 2.52 | 2.10 | 5.88 | 0.78 | 1.64 | 4.17 | 4.20 |
| NAOOjb | 2.94 | 1.02 | 9.51 | 2.63 | 2.07 | 5.63 | 1.04 | 1.61 | 4.28 | 4.38 |
| GOT4.10 | 2.80 | 0.70 | 8.54 | 2.77 | 2.43 | 5.72 | 0.81 | 1.61 | 4.04 | 3.83 |
| HAMTIDE12 | 2.63 | 0.62 | 8.60 | 2.59 | 1.97 | 5.53 | 0.78 | 1.58 | 3.96 | 3.95 |
| EOT20 | 11.18 | 0.75 | 9.32 | 2.61 | 2.37 | 5.57 | 2.07 | 1.49 | 5.73 | 4.96 |
| FES2014 | 1.98 | 0.96 | 9.21 | 2.58 | 2.07 | 5.88 | 0.96 | 1.67 | 4.17 | 4.31 |

*3.4. MSS*

It is essential to define the local MSS/sea surface between the coastal ATGs and altimeter footprints in the Cal/Val campaigns [12,15]. The MSS corrections in altimeter calibration can be maintained by self-measured sea surface and global/regional MSS models [18,33]. The MSS is the sum of the marine geoid and mean dynamic topography (MDT), in which the geoid dominates. The MSS gradient is small over most of the oceans. However, in the vicinity of continental shelves, trenches, islands, sea mounts, and coastal zones with significant terrain changes, such gradients can exceed up to 10 cm/km [33–36].

### 3.4.1. Campaigns of Sea Surface Measurements

Two campaigns have been conducted in 2018 and 2022 to measure the sea surface between the ATG of Zhi'wan island and the footprints of the HY-2B. In 2018, a GPS towing-body was used to measure the sea surface of this area with a width of 6 km and about 26 km from the southeast of Zhi'wan island (Figure 5a). The temporal GNSS reference station was placed on Zhi'wan island to process the kinematic towing-body data. The precision of the GPS towing-body was evaluated using the DGB. The gradient of the measured sea surface is about 1.62 cm/km along the HY-2 orbit, which has a bias of −5~5 cm compared with CNES_CLS11 MSS [18].

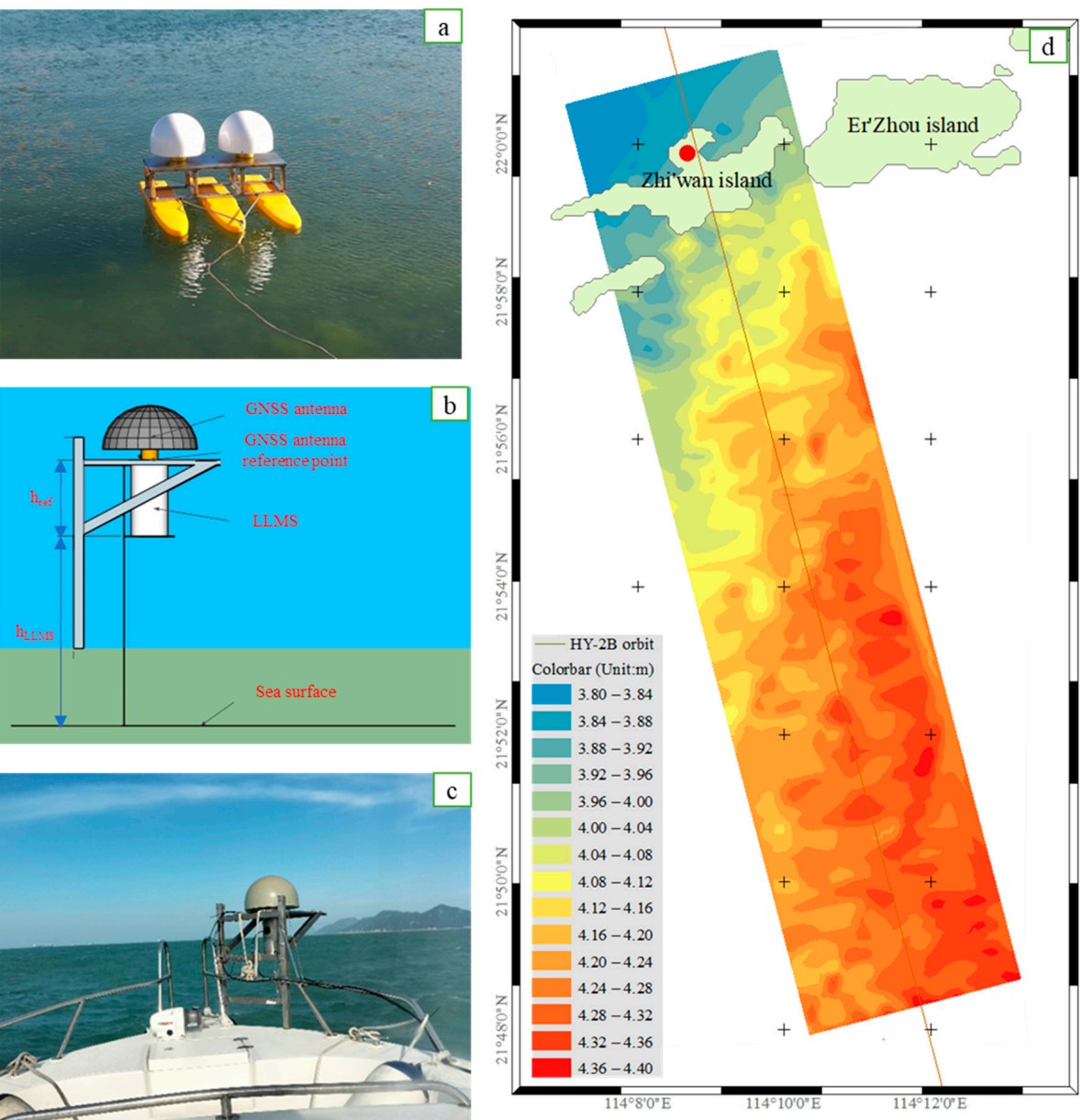

**Figure 5.** Sea surface measurements along the ATG of Zhi'wan island and HY-2B orbits. Image (**a**) is the GPS towing-body used in the campaign of 2018. Image (**b**) is the principle of SSH measurements for HASMS. Image (**c**) is the displacement of the HASMS and the data acquisition and the GNSS receiver were placed in the cabin. Image (**d**) is the measured sea surface of this area using the ordinary kriging interpolation of the measured lines in ArcGIS 10.4.

In 2022, an HASMS developed by the National Ocean Technology Center (NOTC) was used to measure the sea surface of this area. This system consists of a geodetic type GNSS receiver and Chock ring antenna, a liquids level measurement sensor (LLMS), and a data acquisition module (Figure 5b). It was installed at the front of the ship in the campaign (Figure 5c). The SSH measured by the HASMS ($SSH_{HASMS}$) can be estimated by Equation (7):

$$SSH_{HASMS} = H_{GNSSref} - h_{ref} - h_{LLMS} \tag{7}$$

where the $H_{GNSS\,ref}$ is the height measured by the Chock ring antenna;
$h_{ref}$ is the height from the Chock ring antenna reference point to the LLMS, which was measured within 1 mm;
$h_{LLMS}$ is the height from the LLMS to the sea surface.

The NMEA data of the GNSS receiver was collected together with the LLMS to keep the time consistency with an interval of 1 s. An experiment was also undertaken to evaluate the accuracy and precision of HASMS using the DGB and a pressure tide gauge. The pressure tide gauge was placed on the wharf, and the HASMS was moored less than 100 m together with the DGB. The bias was 1.33 mm compared between the HASMS and DGB [17]. The standard deviations were 12.73 mm and 36.66 mm compared by the HASMS with the DGB and the pressure tide gauge, respectively (Figure 6). The SSH measured by the HASMS and the DGB show very good consistency after eliminating the influence of the ocean [17].

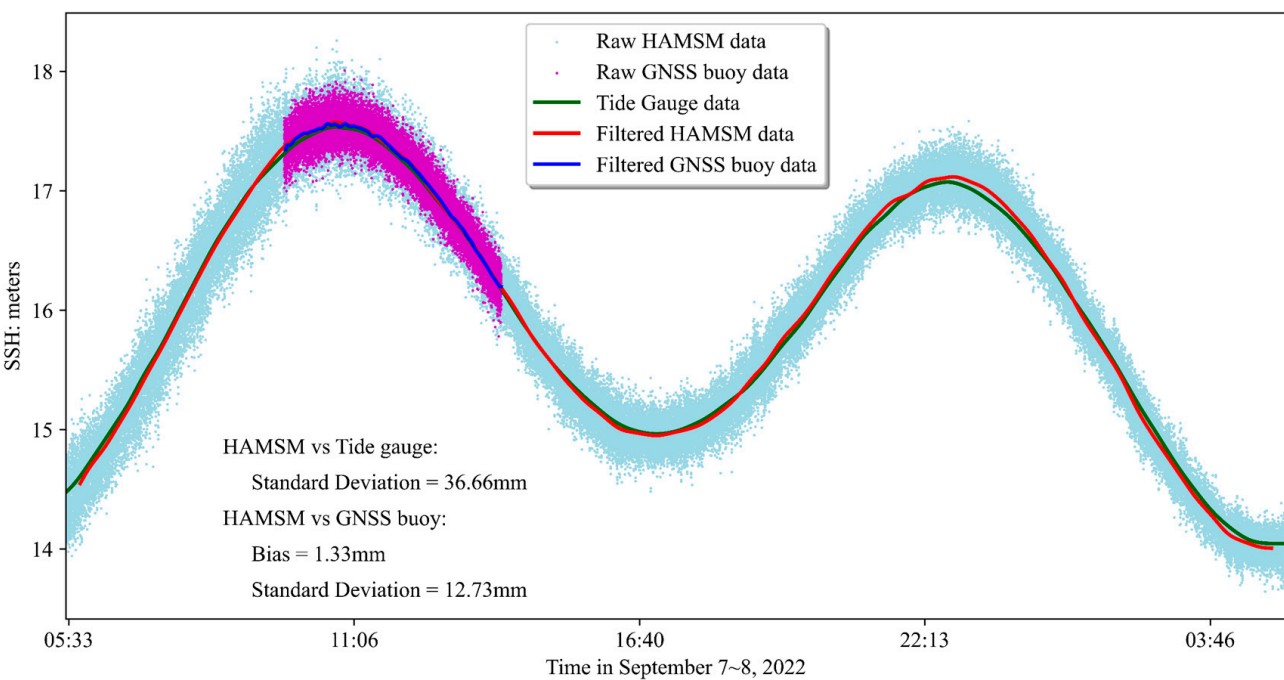

**Figure 6.** Comparisons between the DGB and the HAMSM. The distance between the two equipment is less than 100 m.

The measured sea surface in the two campaigns was interpolated using the cubic interpolation method. It also includes the effects of average sea level, atmospheric pressure, sea state deviation, etc. Therefore, the measured sea surface is the result of a combination of multiple factors.

The sea surface of this area is low in the northwest and high in the southeast, with a difference of about 1.05 cm/km along the orbit of HY-2B. A comparison has been made between the measured sea surface and the global MSS models such as the DTU 2021 (www.space.dtu.dk, accessed on 18 February 2023) [34], EGM 2008 (http://earth-info.nima.mil, accessed on 19 May 2019) plus CLS_MDT_2018 (ftp://ftp-access.aviso.altimetry.fr,

accessed on 28 July 2019) [35,36], and CLS_MSS2015 (ftp://ftp-access.aviso.altimetry.fr, accessed on 28 July 2019) [37]. The standard deviations were 2.34 cm, 2.26 cm, and 2.39 cm, respectively (Figure 7). This indicates good measurement results and fully demonstrates the changes in the local sea surface. The measured sea surface was used for the Cal/Val of HY-2B altimeter data.

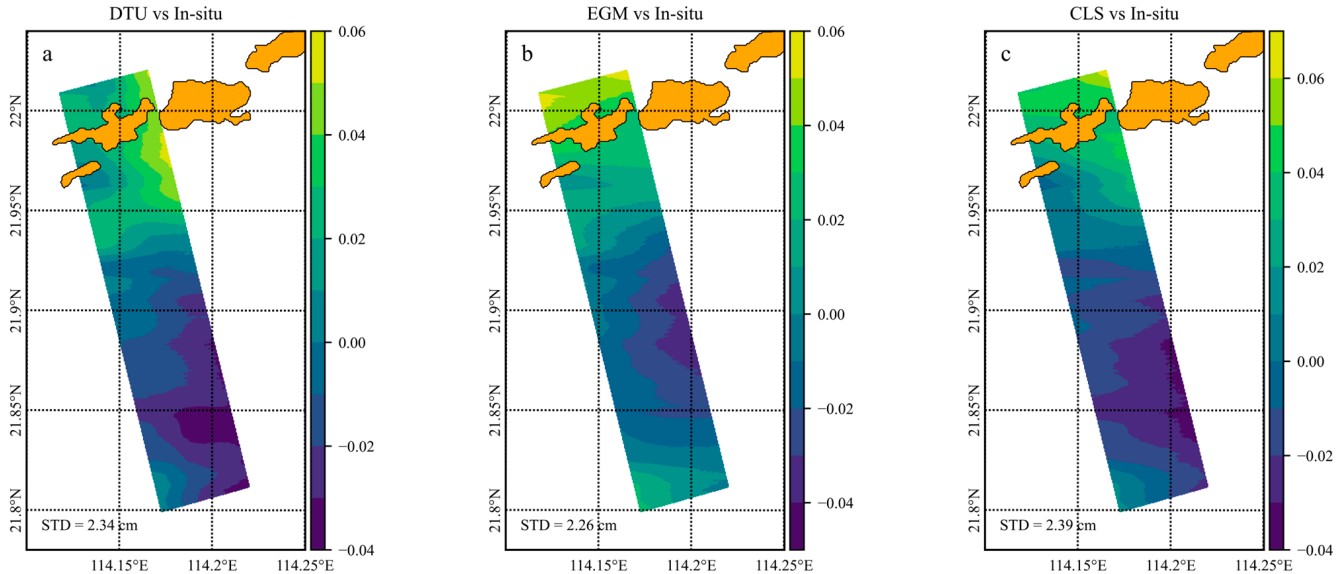

**Figure 7.** Error distributions between the in situ sea surface measurements and DTU 2021 (**a**), EGM2008 plus CLS_MDT_2018 (**b**), and CLS 2015 (**c**).

3.4.2. Mean Sea Surface Models

The above in situ sea surface can make connections between the HY-2B footprints and the ATG of Zhi'wan island in the Cal/Val of altimeters. However, there are no such models for other altimeter calibrations (Figure 2). Therefore, the global MSS models of DTU2021, CLS2015, and EGM2008 subtracted by CLS_MDT_2018 (EGM_CLS) were used.

It is a difficult problem to validate the reliability and accuracy of MSS models, as altimeter provides the most accurate SSH measurements and nearly all available altimeter data have already been used in the derivation of the MSS [37,38]. Two methods were used to validate the precision of the three MSS models in WSCS (113.5~115°E, 21.5~22.5°N). Firstly, we make comparisons between the three models. Results showed that the standard deviation were 6.78 cm, 3.74 cm, and 6.24 cm of CLS2015 vs. DTU2021, CLS2015 vs. EGM, and DTU2021 vs. EGM_CLS, respectively (Figure 7). The MSS accuracy of CLS2015 and EGM_CLS is equivalent. Secondly, the HY-2A altimeter (GDR product derived from NSOAS in 2018), which is not fused in the derivation of the MSS, was used to evaluate the precision of the models. This satellite launched on August, 2011 by NSOAS of China, in order to measure the SSH, WS, sea surface temperature, SWH, etc. [39]. The standard deviation of the SSH of HY-2A was ~7.0 cm compared with the Jason-2 data, which showed good accuracy for validation of MSS models (Figure 8) [39]. The descending pass No. 190 and ascending pass No. 203 flew over this area (red lines in Figure 8). The SSH of the altimeter was derived using the methods in Section 2.2. The standard deviations between HY-2A and DTU2021, CLS2015, and EGM_CLS were 2.14 cm, 2.08 cm, and 2.20 cm, respectively.

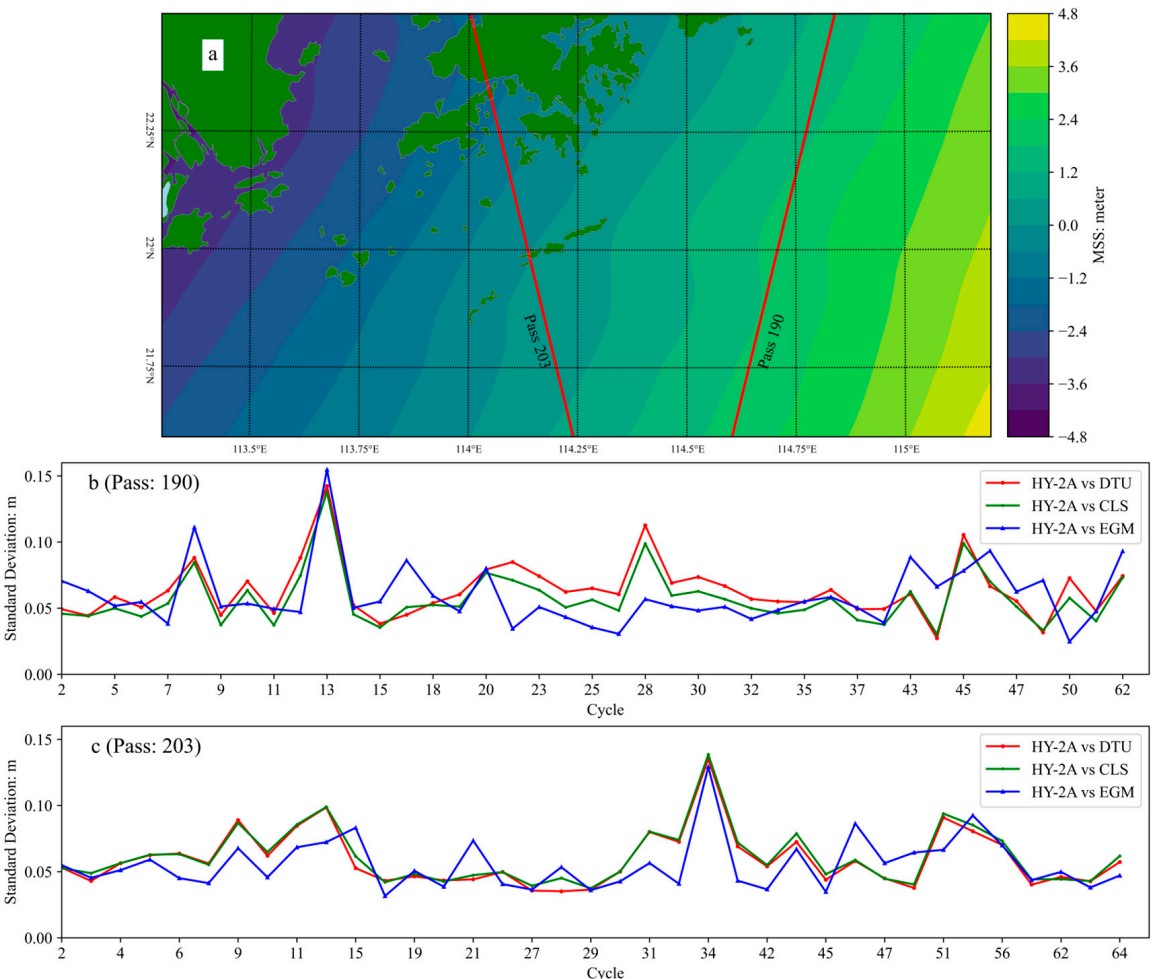

**Figure 8.** MSS of the WSCS. Image (**a**) shows the orbits of HY-2A (red lines). Images (**b**,**c**) are the comparisons between the HY-2A SSH of Pass No. 190 and 203 and MSS of DTU2021, CLS2015, and EGM (MDT minus by CLS_MDT_2018), respectively.

## 4. Calibration Results

### 4.1. HY-2B

The HY-2B satellite launched on 25 October 2018, which was the second altimeter satellite of China and operated by NSOAS. The orbit of the HY-2B satellite maintains a 14-day period for oceanic applications with an inclination of 99.34° and an altitude of about 970 km. The ascending Pass No. 375 of HY-2B flies over the WSCS on Zhi'wan island.

The SSHs of the 64, 65, 94, and 110 cycles of Pass No. 375 were shown as an example of the SSH of HY-2B when it flies over WSCS (Figure 9a). In all these cycles, the SSHs of 20 Hz data were distorted about 10 km from the island (about 21.9°N) when the satellite flies over the WSCS on Zhi'wan island (the green x dots in Figure 9a), especially the 65 and 94 cycles. Therefore, the Cal/Val activities were shown using the data about 10~25 km south of the island. However, the SSH returns to normal after crossing the Zhi'wan island, as there is the presence of a large sea area which is about 30 km, until it flies over the islands of Hong Kong. There are 2~4 footprints of HY-2B and about 25 km between Zhi'wan and Wai'ling'ding islands (Figure 2), which can be used for the Cal/Val of the altimeter.

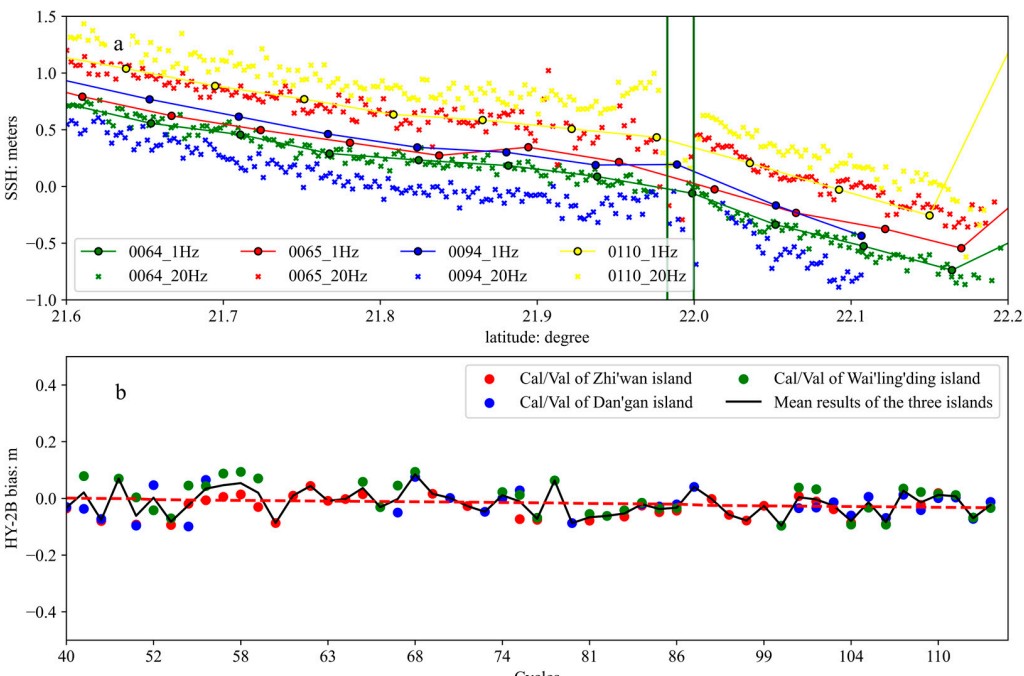

**Figure 9.** The SSH of HY-2B when it flies over the Zhi'wan island and the calibration results. Image (**a**) represents the SSH of HY-2B. The dark green lines represent the Zhi'wan island. The x dots represent the HY-2B SSH of 20 Hz data, and the circles represent the 1 Hz data. Image (**b**) shows the calibration results using the ATGs of WSCS. The red dotted line in image (**b**) represents the drift of the SSH.

The measured sea surface and HAMTIDE12 tide model were used to maintain the tidal difference between the footprints of the HY-2B altimeter and the ATGs of Zhi'wan island. The Cal/Val result was $-30.7 \pm 39.6$ mm. The Cal/Val results were $-21.7 \pm 44.5$ mm and $1.8 \pm 56.9$ mm for ATGs of Wai'ling'ding and Dan'gan islands, respectively, using the CLS 2015 MSS and HAMTIDE12 tide models. The total bias was $-16.7 \pm 45.2$ mm, with a drift of 0.5 mm/year (Figure 9b). The bias of HY-2B SSH is in good agreement between the WSCS and the dedicated sites in Gavdos, Qianliyan, and Zhimaowan [3,15].

### 4.2. HY-2C

The HY-2C was launched on 21 September 2020, which is the third ocean dynamic environment monitoring satellite after HY-2A and HY-2B and the second satellite in the marine dynamic satellite series established as part of China's national civil space infrastructure [40].

The ascending Pass No. 170 and descending Pass No. 185 fly over the Dan'gan island in WSCS (Figure 2). The two passes of 80 and 81 cycles were shown as an example of the SSH measurements (Figure 10a). In the two cycles, the SSH of the altimeter was good when the footprint was 5 km from the southeast of Dan'gan island. Considering the effect of contamination of the island to the microwave radiometers [10], the Cal/Val campaign has been conducted in the area from 10 to 25 km using the ATGs of Dan'gan and Zhi'wan island (on the southeast of Dan'gan island). There was also an open sea area for HY-2C after the overflight of Dan'gan island. The SSH of such an area was less contaminated by the land/island. Therefore, the ATG of Wai'ling'ding island was also used for the Cal/Val in both passes in a distance of about 20 km from the footprint of the altimeter (Figure 2).

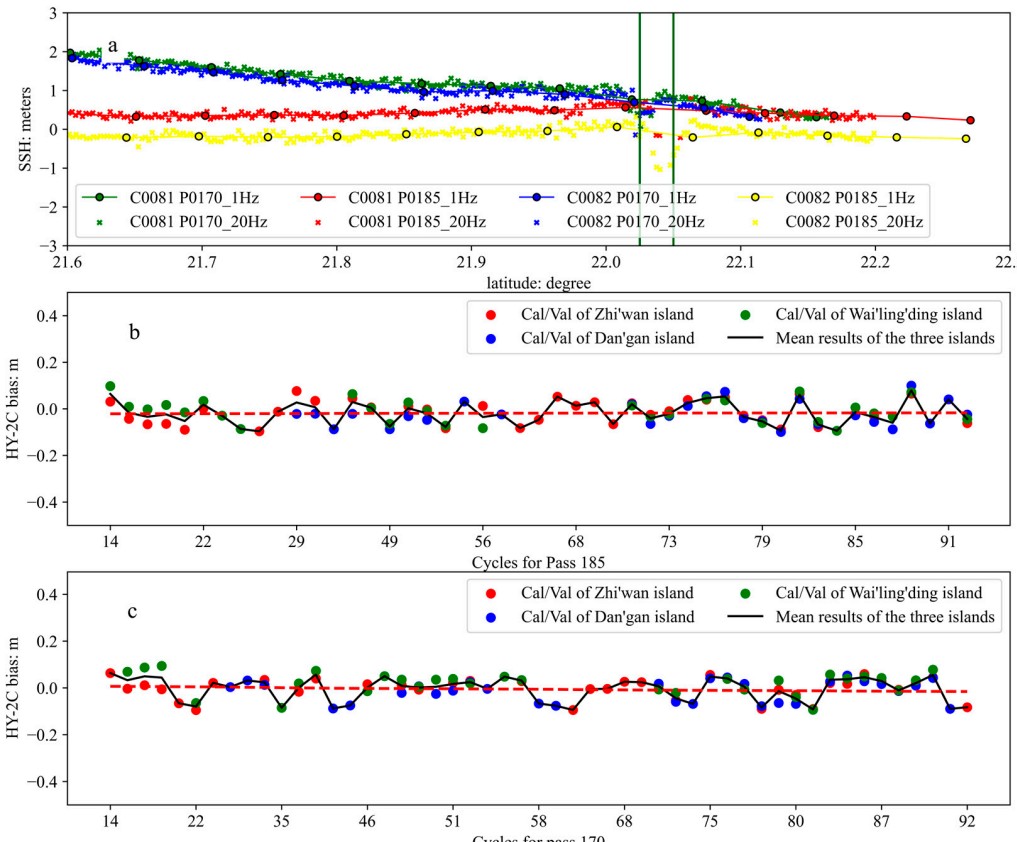

**Figure 10.** The SSH and the calibration results of HY-2C when it flies over the Dan'gan island. Image (**a**) represents the SSH results of the altimeter when it flies over WSCS. The dark green lines represent the Zhi'wan island. The x dots represent the HY-2B SSH of 20 Hz data, and the circles represent the 1 Hz data. Images (**b**,**c**) were the calibration results for Pass No. 185 and Pass 170, respectively. The red dotted lines in image (**b**,**c**) represent the drifts of the SSH.

The CLS 2015 MSS model and HAMTIDE12 tide model were used for MSS and tidal correction as described in Sections 3.3 and 3.4. The mean biases have been determined to be $-18.9 \pm 48.0$ mm for Pass No. 170 and $-5.6 \pm 49.3$ mm for No. Pass 185, with drifts of 0.0 mm/year and $-0.3$ mm/year, respectively. Such results are consistent with each other for different passes.

*4.3. Jason-3*

Jason-3, which was launched on 17 January 2016, is the successor to the Jason-2, Jason-1, and T/P missions. These series of satellites measured SSH, WS, and SWH since 1992. The mission of these satellites is to provide high quality and global view measurements for ocean science and operational products related to climate change studies, sea level rise, ocean circulation, etc. Jason-3 is an international cooperative satellite altimeter mission between the NASA, National Oceanic and Atmospheric Administration (NOAA), EUMETSAT, and CNES. The ascending Pass No. 153 (before April 2022) and descending Pass No. 012 (after April 2022) flies over the WSCS, which makes it possible to assess the performance of the altimeter. In the Cal/Val of GDR data of Jason-3 altimeter, the CLS 2015 MSS model and HAMTIDE12 tide model were used to maintain the MSS and tidal corrections.

In WSCS, the strike of Dan'gan-Zhi'wan-Miao'wan islands formed an island chain. The southeast direction of the island chain is the open ocean. Before April 2022, the orbit of Jason-3 follows in the footsteps of the TOPEX/POSEIDON, Jason-1, and Jason-2 missions (Figure 11a). The Pass No. 153 of Jason-3 orbit crosses some small islands of the strike and to Wai'ling'ding island (Figure 11c). The bias has been estimated in the area less than 25 km from the ATGs. There are three points for 1 Hz of data of the Jason-3 altimeter in such an

area and the SSH performed well in the middle point, which is less contaminated by the island and will be used in the Cal/Val. The altimeter has been calibrated using the ATGs of Zhi'wan and Wai'ling'ding islands, and the bias was determined to be $-4.1 \pm 78.7$ mm (Figure 11d).

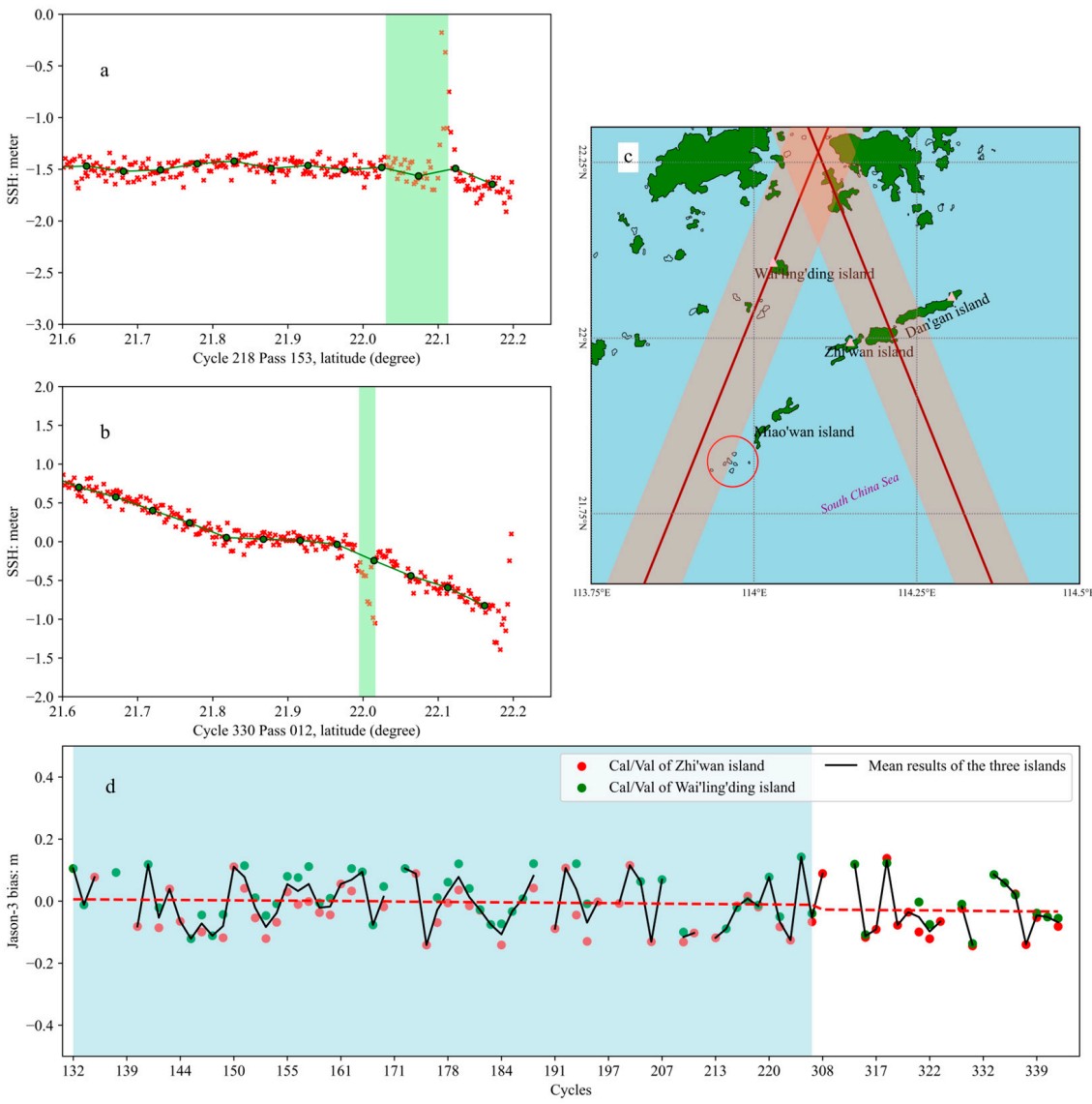

**Figure 11.** The SSH of Jason-3 when it flies over the WSCS. Images (**a**,**b**) are examples of the SSH before and after cycle 301 (April 2022). The red x and green dots in images (**a**,**b**) represent the 20Hz and 1Hz SSH data, respectively. The light green areas represent the small islands of the strike to Wai'ling'ding island and Zhi'wan island, respectively, which may contaminate the SSH. Image (**c**) shows the orbits of the two passes. There are about 6 small islands in the red circle that may contaminate the SSH or wet zenith delay of the altimeter. Image (**d**) shows the bias and drift (red line) of the altimeter and the light blue area represents the cycles before 301 (Pass No. 153). The red dotted line in image (**d**) represents the drift of the SSH.

After April 2022 (Cycle 301), the Jason-3 changed its orbit and the Pass No. 012 crossed over the WSCS on Er'zhou island, which is about 1 km northeast of Zhi'wan island (Figure 2). The ATGs of Zhi'wan and Dan'gan islands were used to calibrate the SSH south of the islands, south of which was the open sea area, where the land contamination was small. The ATG of Wai'ling'ding island was also used to calibrate the SSH of about 10~20 km from its

location (Figure 11b). All the ATGs of the three islands were used for the calibration. The bias of the Pass No. 012 was −25.8 mm, with a standard bias of 85.5 mm (Figure 11d).

During 2019 to 2023, there appears to be no obvious drift in the WSCS (about −0.1 mm/year). Although data were from different passes, the performance of the altimeter in the WSCS is displayed.

### 4.4. Sentinel-3A

S3A is part of the Sentinels constellation of the Copernicus program. It has been in orbit since February 2016 with an orbital cycle of 27 days (14 + 7/27 orbits per day, 385 orbits per cycle) [7]. The synthetic aperture radar (SAR) mode Radar Altimeter (S3A-SRAL) of S3A helps to provide better measurements of coastal areas. The S3A-SRAL was improved along track resolution (approximately 250 m) and, in SAR mode, facilitates SSH measurement close to the coast. S3A has two orbits in this area, of which the descending Pass No. 260 flies over Zhi'wan and Er'zhou islands, and the ascending Pass No. 309 flies over Dangan island (Figures 2 and 12a).

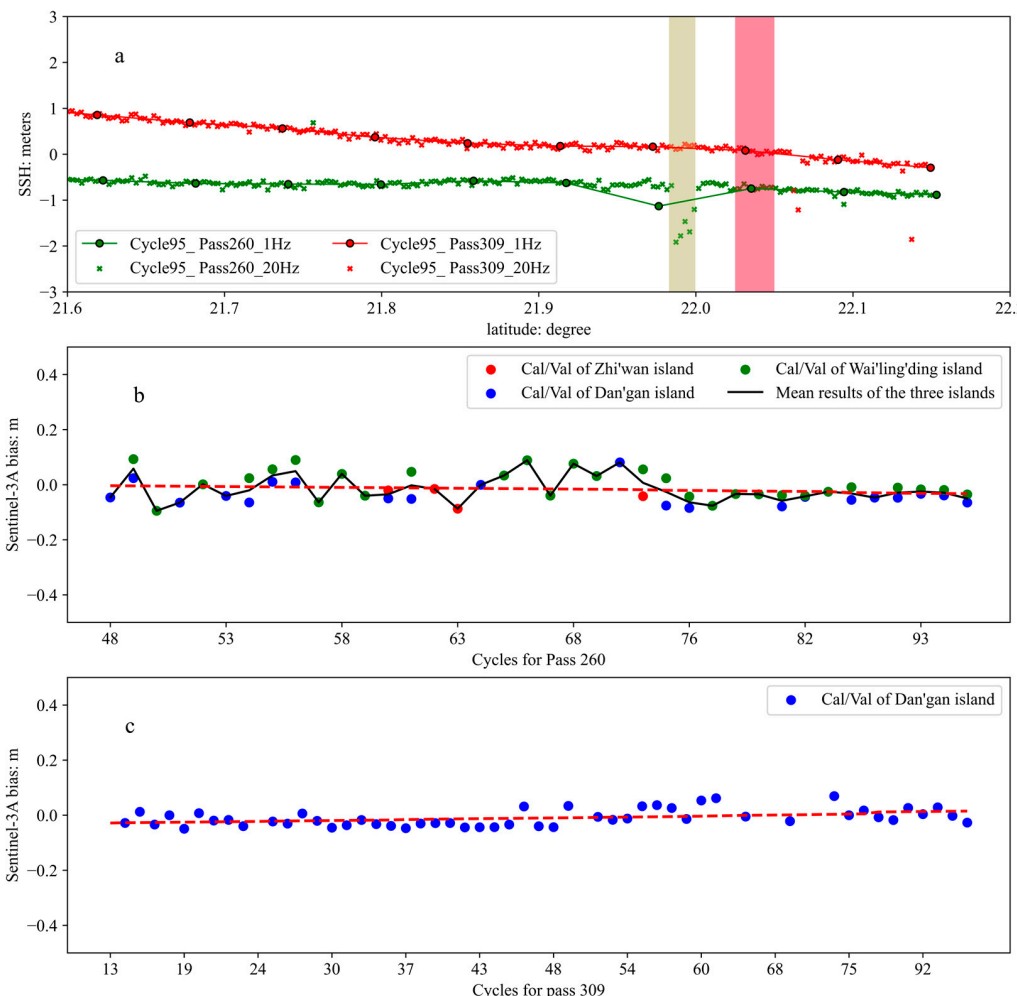

**Figure 12.** The SSH and the bias of S3A when it flies over the WSCS. Image (**a**) shows the example of S3A flying through the WSCS. The red dots and magenta-filled area represent the SSH of Pass No. 260 and the Zhi'wan island. The green dots and khaki-filled area represent the SSH of Pass No. 309 and the Dan'gan island. Image (**b**) shows the bias of S3A Pass No. 260 calibrated by the ATGs of the three islands of WSCS, and (**c**) shows the biases of Pass No. 309 calibrated by the hydrological station and ATG of Dan'gan island. The red dotted lines in image (**b**,**c**) represent the drifts of the SSH.

The SSH of S3A altimeter was calibrated using the ATGs of WSCS. The Jason-3, the MSS, and tidal differences were determined from the CLS 2015 MSS model and HAMTIDE12 tide model. The SSH was less contaminated by the islands of WSCS than HY-2B, HY-2C, and Jason-3 as the SAR mode was used. The descending Pass No. 260 was calibrated using the ATGs of Dan'gan, Zhi'wan, and Wai'ling'ding islands. However, the ascending Pass No. 309 was calibrated only using the ATG of Dan'gan island, as the other two islands were too far from the footprints of the altimeter. The sea level data of the hydrological station placed on Dan'gan island were also used in the Cal/Val of the altimeter from January 2017 to April 2020.

The absolute bias was determined to be $-16.5 \pm 46.3$ mm with a drift of -0.6 mm/year and $-9.8 \pm 30.1$ mm (Figure 12b) with a drift of 0.5 mm/year (Figure 12c) for Pass No. 260 and 309, respectively. The results showed that the SSH of S3A displays a negative sign for the two calibrating passes. Such results are equivalent to the bias calibration in [7].

## 5. Discussion

The WSCS has been in trial operation since August 2019 and has been in continuous operation so far. At the beginning of the construction of WSCS, it was designed for the Cal/Val of the HY-2 series altimeters. Fortunately, it still has the potential to calibrate other altimeters. In previous experiments, we completed the calibration of Jason-3 and S3A satellites using temporary GNSS reference stations, DGB, and tide gauge stations, and the calibration results were basically consistent with those in other calibration sites [10].

At present, we calibrate the HY-2B, HY-2C, Jason-3, and S3A satellite altimeters using the ATGs placed on the islands of WSCS. The datum of the ATGs was regularly transmitted from the PGSs to the sensors during 2019 to 2023. The MSS and tide models used in this paper are global/regional models, and their accuracy is limited. We plan to use a combination of in-site data and remote sensing data to construct a regional MSS/sea surface and tidal model, in order to improve the calibration accuracy of satellite altimeters. The wet atmospheric delay of the altimeter is also an important correction for its distance measurement, using PGSs and GPS radiosonde data [1,5,10].

The DGB together with the PGSs or temporary GNSS stations were also used for Cal/Val of HY-2B and HY-2C altimeters at WSCS. The DGB was moored less than 0.5 km from the altimeter footprints, and more than 2 h before and after the altimeter overflights. The baseline length between the PGSs and the DGB was less than 20 km, which could measure the SSH in an accuracy of less than 35 mm [17]. The HY-2B and HY-2C altimeters were calibrated for six and four times during the on-orbit test campaign in 2018 and 2020, respectively. The estimated accuracy of the HY-2B altimeter's SSH was a 21.2 mm bias with a standard bias of 20.8 mm, which was equivalent to the calibration results of the ATGs. The standard bias of the Cal/Val results was 7.9 mm, which demonstrated high precision of altimeter measurements.

At the beginning of the founding of WSCS, three methods were planned to accomplish the Cal/Val of the altimeters, including the ATGs, moored GNSS buoys (MGB), and bottom pressure tide gauges (BPTG). The three methods complement and test each other and improve the accuracy of the Cal/Val of altimeters. This is incomparable to other international calibration sites. In addition to existing PGSs, ATGs, and meteorological equipment, there are also other planned equipment for the Cal/Val of WSCS, such as the MGB and BPTG, which will be placed in a timely manner. These devices will be directly placed at the footprint HY-2 altimeters (Figure 13), and the datum will be defined using the PGSs and DGB of WSCS. Such devices will ignore the MSS and tide model corrections and will also be used for SWH and WS measurement, thereby completing the validation of such products of the altimeter.

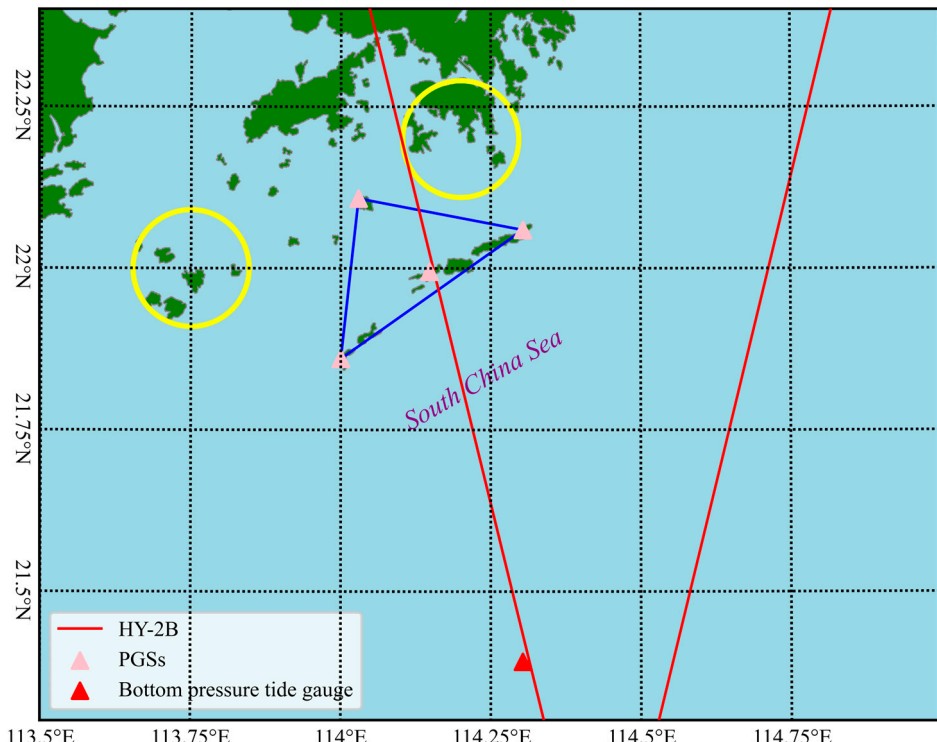

**Figure 13.** The planning devices (red triangle) and the area of WSCS (blue lines). The planning devices include the moored GNSS buoy, the bottom pressure tide gauge, and the hydro-meteorological buoy, which will be placed at the footprint of HY-2B altimeter. The blue lines represent the area of WSCS, which can be used for the Cal/Val of a wide swath altimeter such as SWOT.

In the future, it is necessary to establish more calibration sites to carry out Cal/Val for other satellite altimeters such as HY-2D and SWOT. More calibration verification campaigns for the equipment of the calibration sites should be conducted to ensure consistency in the benchmark, deviation, offset, and other information. Moreover, the connection of several islands in WSCS (blue lines in Figure 13) is similar to a triangle. If another station is set up near Hong Kong or the Da'wan'shan island area (yellow circles in Figure 13), it can achieve calibration of the wide swath satellite altimeter such as SWOT and Guanlan missions.

## 6. Conclusions

From August 2019 to June 2023, the WSCS has been in operation for over three years now and has completed multiple satellite calibrations [3,15]. Two PGSs of Zhi'wan and Wai'ling'ding islands were processed using the GAMIT/GLOBK and Hector software to analyze the trend of this area, especially in U direction. The datum of the three ATGs in WSCS were measured using the DGB each year, to guarantee the same datum with the altimeters after translation. The sea surface was measured to make connections between HY-2B altimeter footprints and the ATG of Zhi'wan island. Moreover, peripheral data such as the sea level of B319 and the hydrological station data (Dan'gan island) and the adjacent GNSS data were also used for the calibrations. The global MSS and tide models were used to maintain the MSS and tidal corrections between the ATGs and the altimeter footprints. The above data and technology support the Cal/Val of WSCS altimeters in this research.

The series of islands on the southwest side of the WSCS have a certain impact on the SSH product of the satellite altimeter. The SSH of the satellite altimeter was good at the sub satellite point about 15 km away from these islands. Among them, S3A satellite altimeter data is the best, and its SAR mode can monitor sea level change information of about 350 m.

From August 2019 to April 2023, the SSH of HY-2B, HY-2C, Jason-3, and S3A was calibrated using the facilities of WSCS. The mean bias of HY-2B was -16.7 $\pm$ 45.2 mm, with

a drift of 0.5 mm/year. The HY-2C biases were −18.9 ± 48.0 mm for Pass No. 170 with drifts of 0.0 mm/year and −5.6 ± 49.3 mm for Pass No. 185 with −0.3 mm/year drifts, respectively. The Jason-3 altimeters changed their orbits on April 2022 and the biases were −4.1 ± 78.7 mm and −25.8 ± 85.5mm for Pass No. 153 and 012, respectively. The absolute biases of S3A were determined to be −16.5 ± 46.3 mm with a drift of −0.6 mm/year and −9.8 ± 30.1 mm with a drift of 0.5 mm/year for Pass No. 260 and 309, respectively.

The function of the calibration sites for satellite altimeters is to unify different satellite altimeters to the same datum, thereby ensuring the continuity of satellite altimeter measurements. This continuity has been maintained for over 30 years [1,5,6]. The construction and professional operation of the WSCS will inject new strength into the calibration of satellite altimeters in the future, providing data for the global Cal/Val of altimeter measurements. In the future, new altimeter calibration points and higher precision regional MSS and tide models will be established to ensure the uniformity in the Cal/Val of multi-source satellite altimeters.

**Author Contributions:** Methodology, W.Z. (Wanlin Zhai), J.Z., H.P., C.C., L.Y., H.W., W.Z. (Wu Zhou) and H.G.; Resources, H.P., C.C., H.W., X.H., W.Z. (Wu Zhou), H.G. and Y.Z.; Data curation, W.Z. (Wanlin Zhai), H.P., C.C., L.Y., X.H., W.Z. (Wu Zhou), H.G. and Y.Z.; Writing—original draft, W.Z. (Wanlin Zhai); Writing—review & editing, C.C. and L.Y.; Funding acquisition, J.Z. All authors have read and agreed to the published version of the manuscript.

**Funding:** This research was funded by National Key Research and Development Program of China, grant number 2021YFC2803304; Hainan Province Science and Technology Special Fund, grant number ATIC-202301003; Shandong Province for Pilot National Laboratory for Marine Science and Technology (Qingdao), grant number LSKJ202201302, 2019B02; National Satellite Ocean Application Service, grant number G623CY125.

**Data Availability Statement:** The PGS and ATG data of WSCS are available from ftp://1.203.103.214 (accessed on 28 August 2023), and the HY-2B and HY-2C data are obtained from NSOAS (https://osdds.nsoas.org.cn/OceanDynamics, accessed on 28 August 2023) with the permission of NSOAS. The PGS data near WSCS are obtained from GRC and are available from ftp://igs.gnsswhu.cn (accessed on 28 August 2023). The Jason-3 (accessed on 28 August 2023), MSS, FES2014 tide model (accessed on 6 July 2023), CLS_MSS2015 and CLS_MDT_2018 (accessed on 28 July 2019) data are obtained from AVISO and are available from ftp://ftp-access.aviso.altimetry.fr/geophysical-datarecord/jason-3. The tide station data of Hong Kong (No. B329) is derived from GLOSS and is available from http://uhslc.soest.hawaii.edu/data/ (accessed on 19 July 2023). The HAMTIDE 12 (accessed on 18 February 2023), EOT20 (accessed on 9 July 2023), DTU16 (accessed on 28 August 2023), NAO99Jb (accessed on 4 September 2019) and GOT4.10 (accessed on 9 July 2023) tide model data are obtained from ftp://ftp-icdc.cen.uni-hamburg.de/hamtide/, https://www.seanoe.org/data/00683/79489/, ftp://ftp.space.dtu.dk, https://www.miz.nao.ac.jp/staffs/nao99/index_En.html and https://earth.gsfc.nasa.gov/geo/data/ocean-tide-models, respectively. The DTU 2021 global MSS model (accessed on 18 February 2023) and EGM 2008 (accessed on 19 May 2019) are available from www.space.dtu.dk and http://earth-info.nima.mil, respectively.

**Acknowledgments:** The authors would like to extend their sincere gratitude to Peng lin, Da Zhao, Zhenfeng Lai and Huanhuan Ding from Guangdong Sea Star Ocean Sci. and Tech. Co., Ltd. We thank NSOAS, AVISO, MIT, IGS and the University of Wyoming for providing the relevant data. We thank the MIT group in the US for providing the GAMIT/GLOBK software available. We are also grateful for the code of Hetcor software provided by http://segal.ubi.pt/hector/. Figures in this paper are plotted with the python software.

**Conflicts of Interest:** The authors declare no conflict of interest.

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
