# Peer review of "Altimeter Calibrations in the Preliminary Four Years’ Operation of Wanshan Calibration Site"

_remotesensing, doi:10.3390/rs16061087_

Round 1
Reviewer 1 Report
Comments and Suggestions for Authors
This study undertakes the calibration and validation of multiple altimeters utilizing observations from the Wanshan calibration site. The authors have incorporated a variety of data sources, and the comprehensive comparisons and investigations conducted are commendable. However, the manuscript currently reads more like a technical report than a scientific article. Significant enhancements are necessary to elevate its scientific merit. My comments are as follows:
The scientific motivation behind the study should be more prominently featured. While the calibration and validation of altimeters are important technical tasks, they alone do not constitute a scientific argument. The introduction needs to be expanded to include more scientific context and background information, explaining why this calibration and validation work is significant from a scientific perspective.
The unique characteristics of the Wanshan calibration site (WSCS) should be clarified in comparison to other existing calibration sites. The authors should articulate what distinct advantages or new opportunities WSCS offers for satellite altimeter calibration that are not available at other sites.
The manuscript presents biases and drifts from various satellite products. An analysis of the differences among these drifts would be insightful. Are these differences attributable to the performance of the individual satellites, or are they related to specific orbital characteristics? A deeper investigation into these aspects would greatly enhance the scientific value of the paper.
With the availability of Surface Water and Ocean Topography (SWOT) data, it would be advantageous to consider including this data in the analysis. If feasible, the inclusion of SWOT data could provide a more current perspective and potentially enrich the study's findings.
Overall, while the technical aspects of the paper are solid, a more pronounced scientific framework is required to meet the standards of a scientific article. The paper would benefit greatly from a clearer exposition of the scientific questions being addressed, as well as a more detailed analysis of the results in the context of these questions.
Author Response
We feel great thanks for your professional review work on our article. As you are concerned, there are several problems that need to be addressed. According to your nice suggestions, we have made extensive corrections to our previous manuscript, the detailed corrections are listed below.
The scientific motivation behind the study should be more prominently featured. While the calibration and validation of altimeters are important technical tasks, they alone do not constitute a scientific argument. The introduction needs to be expanded to include more scientific context and background information, explaining why this calibration and validation work is significant from a scientific perspective.
Answer:
Yes, the introduction was expanded. We added more detail of the altimeter calibration method using the land/island tide gauge in line 80~91. The purpose and scientific motivation of writing this paper was also expanded in line 99~106. The calibration and validation of the altimeters is significant and the scientific arguments are as follows:
- To accomplish the Cal/Val of the altimetersin different regions around the world to evaluate their performance comprehensively. This was added in line 50-51 of the paper.
- To provide a unified datum for multiple altimeters, and ensure the consistency and continuity of the measurements by long-term series altimeters. This was added in line 98~100 of the paper.
- As the first satellite altimeter calibration sitein China, the Wanshan Calibration site can evaluate the measurement accuracy of satellite altimeters in the China sea area, providing a unified absolute measurement datum to ensure the consistency and continuity of satellite absolute sea surface height, significant wave height, wind speed and other parameter measurements.
The unique characteristics of the Wanshan calibration site (WSCS) should be clarified in comparison to other existing calibration sites. The authors should articulate what distinct advantages or new opportunities WSCS offers for satellite altimeter calibration that are not available at other sites.
Answer:
Scientists have pointed out that satellite altimeter calibration needs to be conducted in different regions around the world in order to objectively evaluate the overall performance of the altimeter. (Quartly, G.D.; Chen, G.; Nencioli, F.; Morrow, R.; Picot, N. An Overview of Requirements, Procedures and Current Advances in the Calibration/Validation of Radar Altimeters. Remote Sens. 2021, 13, 125. https://doi.org/10.3390/ rs13010125; Bonnefond, P., Haines, B.J., Watson, C. (2011). In situ Absolute Calibration and Validation: A Link from Coastal to Open-Ocean Altimetry. In: Vignudelli, S., Kostianoy, A., Cipollini, P., Benveniste, J. (eds) Coastal Altimetry. Springer, Berlin, Heidelberg. https://doi.org/10.1007/978-3-642-12796-0_11; Xiaojun Dong, Philip Woodworth, Philip Moore and Richard Bingley. Absolute Calibration of the TOPEX/POSEIDON Altimeters using UK Tide Gauges, GPS, and Precise, Local Geoid-Differences. Marine Geodesy, 2002, 25(3), 189-204). So, several in-situ calibration sites were established to accomplish such task. Therefore, as the first commercial satellite altimeter calibration field in the sea area of China, Wanshan calibration field plays an irreplaceable role. At the beginning of the calibration field design, we plan to use three methods to complete the calibration work of the satellite altimeter: 1) tide gauge, 2) bottom pressure tide gauge, 3) moored GNSS buoy. The three calibration methods complement and verify each other, which can effectively improve the accuracy of satellite altimeter calibration. However, due to certain reasons, the equipment for methods 2) and 3) has not yet been deployed. These have all been elaborated in Section 5.
The manuscript presents biases and drifts from various satellite products. An analysis of the differences among these drifts would be insightful. Are these differences attributable to the performance of the individual satellites, or are they related to specific orbital characteristics? A deeper investigation into these aspects would greatly enhance the scientific value of the paper.
Answer:
Various satellite products may produce deviations or deviations, which are caused by differences in the performance and orbit of individual satellites. The height measurement performance of various satellites varies slightly, but the satellite correction values vary greatly, such as dry and wet atmospheric delay, ionospheric delay, sea state deviation, etc. At the same time, it is also related to the orbit characteristics of the satellite. During the flight process of ascending and descending orbits, it is necessary to switch between land and sea. During the switching process, various measurement equipment or data algorithms may change, which leads to slightly different deviations and offsets of the satellite. These questions have been added to the conclusion of Chapter 6.
With the availability of Surface Water and Ocean Topography (SWOT) data, it would be advantageous to consider including this data in the analysis. If feasible, the inclusion of SWOT data could provide a more current perspective and potentially enrich the study's findings.
Answer:
We have reviewed the SWOT satellite data and found that there is not much difference between the related altimeter data and traditional satellite altimeters (download from ftp-access.aviso.altimetry.fr). Perhaps we have not been able to discover its advantages as soon as possible. In addition, we would like to keep the relevant data and combine it with the content written in Chapter 5 for later calibration and verification of the wide swath satellite altimeter. Therefore, the SWOT data was not shown in this paper, and we shall accomplish the Cal/Val of this altimeter in the future.

Reviewer 2 Report
Comments and Suggestions for Authors
The authors described how the satellite altimeters were calibrated and validated using Permanent GNSS Reference Station data, tide gauge and buoy data, and regional tide models. The authors need to simplify the abstract part to briefly describe the main framework of this study. The authors may provide a flowchart in “2. Dataset and Methods” for more clearly summarizing how various data and methods are combined for the calibration and validation. The authors need to explain more on how “Error distributions” in Figure 7. at Line 320 are derived in the text.
Comments on the Quality of English LanguageLine 235 to Line 236: “The precision of the tide models was evaluated by four in-situ tide gauge stations” can be modified to “The precision of the tide models was evaluated by comparing with the water level data from four in-situ tide gauge stations”.
Author Response
We feel great thanks for your professional review work on our article. As you are concerned, there are several problems that need to be addressed. According to your nice suggestions, we have made extensive corrections to our previous manuscript, the detailed corrections are listed below.
The authors described how the satellite altimeters were calibrated and validated using Permanent GNSS Reference Station data, tide gauge and buoy data, and regional tide models. The authors need to simplify the abstract part to briefly describe the main framework of this study. The authors may provide a flowchart in “2. Dataset and Methods” for more clearly summarizing how various data and methods are combined for the calibration and validation. The authors need to explain more on how “Error distributions” in Figure 7. at Line 320 are derived in the text.
Answer:
We have modified the structure of Section 2 and 3 to provide a clearer explanation of the calibration and validation process of the satellite altimeter in this article. Firstly, in Section 2, the Wanshan calibration site of the satellite altimeter and the calibration method used in this article were introduced in detail. Secondly, in Section 3, the observation data and its processing methods in the calibration site are introduced. 1) the GNSS reference stations are used to establish accurate datum for satellite altimeter calibration; 2) the dedicated GNSS buoy was used to transmit the reference of GNSS reference station to the tide gauge station; 3) The mean sea surface model and tidal model are used to transmit the sea surface height measured by the tide gauge locations to the satellite footprints. The calibration and validation of the altimeters was completed through the above work. Please refer to the revised manuscript for detailed information.
Comments on the Quality of English Language
Line 235 to Line 236: “The precision of the tide models was evaluated by four in-situ tide gauge stations” can be modified to “The precision of the tide models was evaluated by comparing with the water level data from four in-situ tide gauge stations”.
Answer:
Yes. This has been corrected in line 281~282 in the revised manuscript.

Reviewer 3 Report
Comments and Suggestions for Authors
The article is about calibrating an altimeter over a fixed time interval.
The title is in line with the topic of the article.
The abstract is comprehensive, but does not give a clear answer as to what results were obtained.
The introduction presents the background of the research work and the need for this work.
There is no information on the results of the noise analysis for GNSS data. Were daily or weekly data used? Were similar values obtained for all 61 stations?
The authors once use the abbreviation GPS and GNSS interchangeably, are they understood to be the same data?
Is the presented accuracy of 0.01 in Table 1 not too high for this type of research as in the article? According to the literature over a three-year period, an accuracy of 0.5mm/year can be obtained (Ihde and Augath, 2001).
Why did the authors choose stations with high horizontal velocity. more than 3cm/year.
In Figure 3, the velocity for the Y coordinate is not linear. Were jumps in the time series analysed? There are at least two trends. (Kowalczyk and Rapinski 2018).
For what purpose was chapter 2.1 implemented?
What does it mean that the HAMTIDE12 Tidal Model is the most accurate?
Are the changes between the different cycles in Figure 8 linear?
Section 4.4, why fix the absolute bias when its error is three times greater than that (line 494)?
In conclusion, the work done is useful for the calibration of satellite altimeters, but the formulation of the article is confusing to the reviewer, chaotic at times. There are a lot of abbreviations and detailed analyses, as well as very general statements. I suggest that the article be restructured by first formulating the scheme of operations, providing the calibration method, testing and use of the data, and collating the calibration results into one readable whole.
In its current form, the article is poorly readable and will certainly make it difficult for the reader to find the results that interest him. It will also make it difficult to popularise these results. The summary is extensive, but does not give a clear answer as to what results were obtained.
Author Response
We feel great thanks for your professional review work on our article. As you are concerned, there are several problems that need to be addressed. According to your nice suggestions, we have made extensive corrections to our previous draft, the detailed corrections are listed below.
The article is about calibrating an altimeter over a fixed time interval.
The title is in line with the topic of the article.
The abstract is comprehensive, but does not give a clear answer as to what results were obtained.
The introduction presents the background of the research work and the need for this work.
Answer:
Yes. The results were given in the new manuscript.
There is no information on the results of the noise analysis for GNSS data. Were daily or weekly data used? Were similar values obtained for all 61 stations?
Answer:
Yes. Noise analysis for GNSS data was not given in this paper. The daily data was used, and this was added in the paper in line 170~172.The GAMIT/GLOBK software was used to process the daily data of the stations, and then the high-precision baseline and coordinate adjustmens were obtained. After that, the noise analysis was done using the Hector software. Five stations was set to be stable in the GAMIT/GLOBK processing. A detailed introduction of this research can be seen in Reference []. (Zhai, W.; Zhu, J.; Lin, M.; Ma, C.; Chen, C.; Huang, X.; Zhang, Y.; Zhou, W.; Wang, H.; Yan, L. GNSS Data Processing and Validation of the Altimeter Zenith Wet Delay around the Wanshan Calibration Site. Remote Sens. 2022, 14, 6235. https://doi.org/ 10.3390/rs14246235)
The authors once use the abbreviation GPS and GNSS interchangeably, are they understood to be the same data?
Answer:
Yes. In this paper, the GPS was used for the GPS towing-body. In this campaign of 2018 (see reference: Zhai, W.; Zhu, J.; Ma, C.; Fan, X.; Yan, L.; Wang, H.; Chen, C. Measurement of the sea surface using a GPS towing-body in Wanshan area. 2020. Acta Oceanol. Sin. 39, 123–132.), only the GPS data was achieved and processed. However, in this paper the Global Navigation Satellite System (GNSS) data (including GPS, BDS, GLONASS, Galileo) was achieved in Wanshan calibration site, although only the GPS was used in the processing. Due to the fact that the calibration field was built in 2019 and the equipment were purchased in 2018, it did not yet support higher precision satellite data from the BeiDou Navigation Satellite System (BDS). However, we are planning to use all the GNSS such as GPS, GLONASS, BDS, Galileo (especially BDS) for comprehensive processing.
Is the presented accuracy of 0.01 in Table 1 not too high for this type of research as in the article? According to the literature over a three-year period, an accuracy of 0.5mm/year can be obtained (Ihde and Augath, 2001).
Answer:
Yes. According to the references(1、Isawi, S., Schuh, H., Männel, B. & Sakic, P. (2022). Stability analysis of the Iraqi GNSS stations. Journal of Applied Geodesy, 16(3), 299-312. https://doi.org/10.1515/jag-2022-0001;2、Saji, A.P.; Sunil, P.S.; Sreejith, K.M.; Gautam, P.K.; Kumar, K.V.; Ponraj, M.; Amirtharaj, S.; Shaju, R.M.; Begum, S.K.; Reddy, C.D.; et al. Surface Deformation and Influence of Hydrological Mass over Himalaya and North India Revealed from a Decade of Continuous GPS and GRACE Observations. J. Geophys. Res. Earth Surf. 2020, 125, 1–17.)。The accuracy of 0.01mm/year is high enough for the global average sea level records .
Why did the authors choose stations with high horizontal velocity. more than 3cm/year.
Answer:
The position of the station is not within our control, as it is closely related to the orbit of the satellite altimeter and the conditions for the construction of the calibration field. However, due to the large horizontal displacement at this location, the vertical displacement is relatively small. And the results measured by the satellite altimeter are mainly in the vertical direction. Based on our long-term tracking and measurement of the GNSS reference station, this calibration field can provide high-precision calibration results. This is also why we need to build a long-term operational benchmark station in the altimeter calibration field. Please refer to the altimeter calibration at the location of the Harvest oil platform in the United States (Haines, B.; Desai, S.; Kubitschek, D.; Leben, R. A brief history of the Harvest experiment: 1989–2019. Adv. Space Res. 2021, 68, 1161–1170.).
In Figure 3, the velocity for the Y coordinate is not linear. Were jumps in the time series analysed? There are at least two trends. (Kowalczyk and Rapinski 2018).
Answer:
The general coordinate time series changes are not linear and have seasonal or annual variations. As this paper only focuses on satellite altimeter calibration, we can use annual variations; The time length of the two GNSS reference stations is relatively short, and the annual changes are not as significant. Therefore, we did not provide a more detailed analysis in our paper.
For what purpose was chapter 2.1 implemented?
Answer:
The destination for Section 2.1 was to introduce the permanent GNSS stations (PGSs) of Wanshan Calibration site, process the GNSS data, achieve the accurate coordinates of the PGSs and then set the datum of the Wanshan Calibration site for altimeter calibration. Such section was modified to 3.1. The tide gauge measures the sea surface with none datum. The datum was defined using the compared measurements such as leveling or comparisons with the GNSS buoy (kinematic solution with the static PGSs data and accurate coordinates).
This datum of the PGSs was WGS 84 ellipsoid with equatorial radius of 6378.137 kilometers and a flattening coefficient of 0.003352811, but have height differences with the S3A, HY-2B, HY-2C and J3 altimeter, which have equatorial radius of 6378.1363 km and a flattening coefficient of 1/298.257. The accuracy of the height datum can reach to less than 5 mm.
What does it mean that the HAMTIDE12 Tidal Model is the most accurate?
Answer:
In the Cal/Val of altimeters, the tide gauge was set on land/island. However, the altimeter footprint was available about 15~30 km from the tide gauge. In such distance, there are mean sea surface and tide difference. So we should maintain such differences using the regional/global models. In this paper, we compared six regional/global tide models and the HAMTIDE12 model was the most accurate. This means that such model can maintain the tidal correction for Cal/Val of altimeters. And this model was used in the Cal/Val of the four altimeters in this paper.
Are the changes between the different cycles in Figure 8 linear?
Answer:
No. They are not linear. The changes in some areas are regular, but in this paper, due to the short measurement time of GNSS reference stations (less than four years), special patterns have not yet been reflected. In the study, we used the calculation results of a single day after removing noise to complete the work in this article.
Section 4.4, why fix the absolute bias when its error is three times greater than that (line 494)?
Answer:
This question confuses us. Due to the calibration of the satellite altimeter, the deviation given is the average value after multiple calibrations (-25.8 mm), while the three times greater data (85.5 mm) is the standard deviation. The fixed absolute bias refers to the overall bias of the satellite altimeter during the calibration period, which is from August 2019 to April 2022 for the Jason-3 satellite.

Round 2
Reviewer 1 Report
Comments and Suggestions for Authors
no more comments
Author Response
We thank the reviewer for the positive comments.
Reviewer 3 Report
Comments and Suggestions for Authors
I accept the improvements made to the article.
Author Response

(The authors gave the same response as above.)
